# An evaluation of bird and bat mortality at wind turbines in the Northeastern United States

**Daniel Y. Choi**[1,2]*, **Thomas W. Wittig**[1], **Bryan M. Kluever**[3]

**1** Division of Migratory Birds, United States Fish and Wildlife Service, United States Department of the Interior, Hadley, Massachusetts, United States of America, **2** Department of Fisheries, Wildlife, and Conservation Biology, North Carolina State University, Raleigh, North Carolina, United States of America, **3** Florida Field Station, National Wildlife Research Center, Wildlife Services, United States Department of Agriculture, Gainesville, Florida, United States of America

* dychoi@ncsu.edu

**Data Availability Statement:** All relevant data are within the paper and its Supporting Information files.

## Abstract

Wind energy offers substantial environmental benefits, but wind facilities can negatively impact wildlife, including birds and bats. Researchers and managers have made major efforts to chronicle bird and bat mortality associated with wind facilities, but few studies have examined the patterns and underlying mechanisms of spatial patterns of fatalities at wind facilities. Understanding the horizontal fall distance between a carcass and the nearest turbine pole is important in designing effective search protocols and estimating total mortality. We explored patterns in taxonomic composition and fall distance of bird and bat carcasses at wind facilities in the Northeastern United States using publicly available data and data submitted to the US Fish and Wildlife Service under scientific collecting and special purpose utility permits for collection and study of migratory birds. Forty-four wind facilities reported 2,039 bird fatalities spanning 128 species and 22 facilities reported 418 bat fatalities spanning five species. Relative to long-distance migratory birds, short-distance migrants were found farther from turbines. Body mass of birds and bats positively influenced fall distance. Turbine size positively influenced fall distance of birds and bats when analyzed collectively and of birds when analyzed separately from bats. This suggests that as turbines increase in size, a greater search radius will be necessary to detect carcasses. Bird and bat fall distance distributions were notably multimodal, but only birds exhibited a high peak near turbine bases, a novel finding we attribute to collisions with turbine poles in addition to blades. This phenomenon varied across bird species, with potential implications for the accuracy of mortality estimates. Although pole collisions for birds is intuitive, this phenomenon has not been formally recognized. This finding may warrant an updated view of turbines as a collision threat to birds because they are a tall structure, and not strictly as a function of their motion.

## Introduction

Despite environmental advantages of wind energy (e.g., renewable resource, near zero carbon dioxide emissions and water requirements [1]) over more traditional, carbon-based energy

**Funding:** The author(s) received no specific funding for this work.

**Competing interests:** The authors have declared that no competing interests exist.

sources, wind facilities can threaten wildlife directly (e.g., collision mortality) and indirectly (e.g., displacement) [2–6]. The magnitude of indirect effects of wind facilities on wildlife are variable. For example, the impacts on elk (*Cervus elaphus* [7]), pronghorn (*Antilocapra americana* [8]), greater prairie chicken (*Tympanuchus cupido* [9]), and desert tortoise (*Gopherus agassizii* [10]) appear limited or benign. However, some species, such as greater sage grouse (*Centrocercus urophasianus*) respond negatively to turbine presence [11]. Negative impacts of wind facilities on wildlife may intensify with the current rapid expansion of the wind industry [12, 13]. For instance, United States (US) wind power has tripled in the past decade [14], and is forecast to more than quadruple by 2050 [1, 15].

To date, the majority of research on the effects of wind facilities on wildlife has focused on mortality of birds and bats (see Schuster et al. [6] for review). Turbine collision mortality in the US has been reported for 300 bird species, with small passerines (Passeriformes) comprising 57% of US fatalities [2]. Recent estimates of annual bird fatalities in North America have ranged between 140,000 and 679,000 [5, 16, 17], though the number of US turbines has increased by more than 35% since those estimates were provided [14]. There is particular concern regarding raptors of low abundance [2, 18, 19], whose low reproductive rates and high adult survival may hamper recovery from mortality at wind facilities [6, 20]. However, for most bird species, wind turbines contribute a small amount to total anthropogenic mortality [21] and may not significantly impact vital rates (e.g., annual survival probability) and state variables (e.g., abundance, density, occupancy) at the population level.

Wind energy development may have greater implications for the conservation of bats. Mortality by turbine collision has been recorded for 24 of North America's 47 bat species, primarily migratory tree-roosting bats [22]. Compared to many passerine birds, wind turbines may have a more pronounced negative impact on migratory tree bats. At the majority of US wind facilities that have been examined, bat mortality has been estimated to be higher than bird mortality [2, 6]. Estimates of annual bat fatalities in North America have ranged between 600,000 and 949,000, but similar to birds, the number of US turbines has increased substantially since these estimates were developed [14]. High bat mortality is of particular concern because basic demographic information is lacking for many species, making population-level inferences of the impacts of wind facilities unfeasible [23, 24]. Further, since 2006, many bats species have been severely impacted by white-nose syndrome and additional mortality from wind facilities could impact recovery of species of special concern [25]. However, it should be noted that migratory tree bats that experience the most mortality at wind facilities are not as affected by white-nose syndrome [26]. Finally, because bats have longer life cycles and lower reproductive rates than many bird species, populations may be less likely to remain stable when faced with additive anthropogenic mortality [22].

Despite research to date, there remains a shortage of information on the spatial arrangement and patterns of carcasses at wind facilities [27]. Understanding what influences fall distance, the horizontal distance between a carcass and the nearest turbine pole, is important in designing effective carcass search protocols (e.g., specifying a minimum search plot radius) [27, 28] and in accurately estimating total mortality [29, 30]. For example, large carcasses have a greater possible fall distance [27] and hence a higher likelihood of being missed during surveys with small search radii (e.g., because of nearby vegetation restrictions) [31]. Thus, accounting for differences in detection probabilities of large and small carcasses is necessary in reliably estimating mortality rates [31, 32]. Past studies have recognized that there may be other factors that influence fall distance in a similar fashion to carcass size, but research remains sparse [29, 30, 33]. For example, the effect of turbine height has been modeled [27] but empirical studies and investigations of the effect of turbine rotor diameter on fall distance are lacking. The effect of animal type (i.e., bird/bat) and carcass size has been both modeled

[27] and empirically shown [22, 34] but these studies have used subjectively defined size categories (large/small) and could be improved upon by using finer size classifications. Fall distance may also vary according to migration behavior (e.g., long/short distance and nocturnal/diurnal), which is known to influence collision risk [2, 29]. A better understanding of the factors affecting fall distance as well as the distribution of fall distances itself will lead to higher quality mortality estimates and a clearer image of the impact of wind facilities on birds and bats [29, 30].

Current estimates of bird and bat mortality at wind facilities are substantially variable due to non-standardized survey methods [16, 32] and inconsistent sampling of wind facilities across the North American continent [2]. Likewise, in some regions, information about carcass species composition is based on data from a relatively small number of wind facilities. For example, Erickson et al. [17] and the American Wind Wildlife Institute (AWWI) [34] summarized species composition for the eastern North American biome based on data from eight and 20 wind facilities, respectively. Each of these studies improved upon past knowledge, but more extensive reviews have been called for to clarify and confirm findings of species composition and collision risk [17, 34]. Many wind facilities report the results of carcass surveys to the US Fish and Wildlife Service (USFWS) in accordance with federal permitting, but much of this data has not been disseminated to the public [16]. In the Northeastern US, 44 wind facilities have reported data to the USFWS, presenting the opportunity for a substantial improvement in our understanding of regional species composition of bird and bat fatalities at wind facilities.

Here, we investigated a dataset of bird and bat fatalities at wind facilities located in the Northeastern US. Our objectives were to: 1) generally describe taxonomic patterns of bird and bat mortality, and 2) examine whether animal type (bird/bat), body size, migration behavior, or turbine size influence the fall distances of carcasses associated with turbines.

## Methods

Between 2008 and 2017, the USFWS received records of individual bird and bat fatalities from a subset of wind facilities in the Northeastern US (Fig 1). Facilities submitted these reports as a condition of either a Migratory Bird Treaty Act special purpose utility permit or scientific collecting permit, which allowed for collection, transport, or possession of migratory birds for mortality monitoring or scientific research and educational purposes, respectively. Fifty percent of permittees voluntarily reported bat fatalities. We are unsure how many of these facilities may have also contributed data to studies such as those by AWWI [22, 34]. However, we include 44 total wind facilities, compared to 20 in the eastern region of North America included by AWWI [34]. Thus, the majority of wind facilities in our study contribute new data in examining species composition.

We supplemented fatality records with classification information and model covariates. By species, we added migration status (i.e., resident, migrant, partial migrant) and population data [35], taxonomic classification information [36, 37], trophic guild information [38], timing (nocturnal/diurnal) and distance (long/short) of migration [39], and body mass [37–39]. By wind facility, we added turbine hub height (distance from the ground to the rotor center) and rotor diameter [14]. We created a second dataset using only fatality records that included fall distance and supplemented it with data from online, publicly available post-construction reports for two additional Northeastern US wind facilities with large sample sizes, Criterion in Maryland [40, 41] and Record Hill in Maine [42].

We used the lme4 package [43] in R [44] to create generalized linear mixed models for fall distance (m) of 1) birds, 2) bats, and collectively, 3) birds and bats (hereafter referred to as the

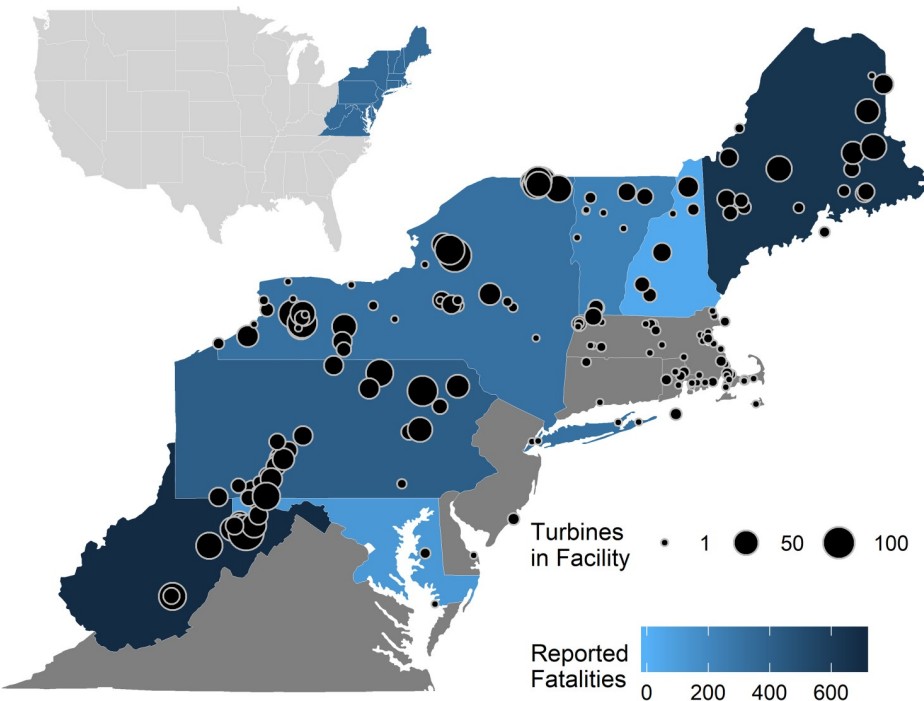

**Fig 1. Cumulative number of bird and bat fatalities reported to the US Fish and Wildlife Service between 2008 and 2017 by 44 wind facilities in the Northeastern US.** States in gray did not report any fatalities. Exact locations of wind facilities that reported data are withheld due to privacy concerns. Black circles represent all wind facilities in the Northeastern US and size indicates the number of wind turbines in each facility [14].

birds-only, bats-only, and all-taxa models). Our predictor variables in these models related to turbine dimensions and animal characteristics (Table 1). All models featured a Poisson probability distribution with a log link and a random effect for wind facility. Prior to model construction, we centered continuous predictors on their means and scaled them by two standard deviations to assist model convergence [45] and enable later comparison of model coefficients [46]. We also rounded fall distances to the nearest integer to meet the Poisson distribution's

**Table 1. Predictor variables used in generalized linear mixed models of bird and bat fall distance.**

| Predictor Variable | Description | Source(s) |
|---|---|---|
| Mass[a,b,c] | For birds, species' male body mass. For bats, mean of species' minimum and maximum body mass | 37, 39, 48 |
| Turbine hub height[a,b,c] | Distance from the ground to the rotor center | 14 |
| Turbine rotor diameter[a,b,c] | Sweep diameter of turbine blades | 14 |
| Turbine diameter: height[a,b,c] | Interaction term between turbine hub height and rotor diameter | 14 |
| Animal type[a] | Bird or bat | NA |
| Migration distance[b] | Long- or short-distance bird migrant | 39 |
| Migration timing[b] | Nocturnal or diurnal bird migrant | 39 |

[a]All-taxa model predictor variable

[b]Birds-only model predictor variable

[c]Bats-only model predictor variable

assumption of non-negative integer values. Where turbine dimensions varied within an individual facility, we assigned to each record the most common hub height and rotor diameter for that facility. We removed fatality records with fall distances greater than 100 m, assuming from previous modeling [27] that these records were likely unrelated to turbine collision. We also removed data that lacked species-specific mass values due to missing or unknown species classification. Lastly, for the birds-only models, we removed records associated with non-migratory species, which lacked migration distance and timing values, therefore violating requirements for later model comparison [47].

Ranking models according to the Information-Theoretic approach [48], we first constructed global models containing all applicable predictor variables (Table 1). We then assessed for multicollinearity among predictor variables, removing variables according to highest variance inflation factor (VIF) [49] until all VIFs were less than 3 [50]. Next, we measured the fit of each global model (birds-only, bats-only, and all-taxa global models) by using the MuMIn package [51] in R to calculate a marginal and conditional pseudo r-squared based on the Tri-gamma-estimate method. We then calculated a dispersion parameter for each global model using the ratio of residual deviance to the residual degrees of freedom. Where overdispersion was evident (i.e., dispersion parameter $> 1$ [52]), we did not attempt an observation level random effect [52], as this would have prevented meaningful interpretation of the marginal and conditional pseudo r-squared values [45].

We then constructed candidate models from all combinations of the global models' predictor variables and ranked them using Akaike's Information Criterion (AIC) or, if the global model was overdispersed, Quasi-AIC (QAIC [53]). We considered candidate models with $\Delta AIC < 6$ as supported by the data [45, 54, 55]. Next, we removed models with uninformative parameters from the model sets according to the procedure outlined by Leroux [56]. In this process, we considered log likelihoods virtually identical when they differed by less than two. We performed model-averaging when multiple models met the $\Delta AIC$ and informative parameter criteria. In averaging top model sets, we used the natural average method over the zero method due to our general interest in all predictor variables and our expectation that some effects might be weak [57]. We considered coefficients as affecting fall distance if their 95% confidence intervals did not overlap zero.

During exploratory analysis, we noted possible multi-modality in the density distributions of bird and bat fall distances. To verify this observation, we performed non-parametric tests using the ACR method [58], as implemented by the multimode package [59] in R. We included all bird and bat records with fall distance information and values of 100 m or less. Our null hypothesis in these tests was unimodality, with $\alpha = 0.05$. We set the lower limits for these tests at 0 m (carcasses with a distance of 0 m were found directly next to the turbine base) to reflect the non-negative nature of distance values. Although data ranged to 100 m, we set our upper limit for testing to 60 m after observing that the ACR method tended to identify modes at the location of single data points in the far right tail of the fall distance distribution when allowed to search out to 100 m. We considered these far-right modes as reflecting nothing more than the method's computational limitations under sparse data conditions. We ran all other function parameters at default values (see [52] and S1 Code). Following these tests, we used the multimode package's implementation of Chaudhuri and Marron's [60] SiZer (SIgnificant ZERo crossings of derivatives) to map the location of significant ($\alpha = 0.05$) increases and decreases in density, and thereby gathered a sense of the potential number and location of modes. Based on these impressions, we estimated the location of modes using the Hall and York [61] method within the multimode package, again specifying the lower and upper limits as 0 and 60 m, respectively.

**Table 2. Five most frequent families of bird mortality at wind turbines in the Northeastern US.**

| Family | Common Name | Number of Fatalities | % of Total Fatalities | % Population Change 1970–2017[a] | % of Species in Decline[a] |
|---|---|---|---|---|---|
| Parulidae | New World Warblers | 576 | 28.25% | -37.60% | 64% |
| Vireonidae | Vireos | 440 | 21.58% | 53.60% | 17% |
| Regulidae | Kinglets | 145 | 7.11% | -7.10% | 50% |
| Phasianidae | Grouse and Allies | 75 | 3.68% | 24.30% | 33% |
| Turdidae | Thrushes | 74 | 3.63% | -10.10% | 55% |

[a]Rosenberg et al. [64]

## Results

### Taxonomic and temporal patterns

Forty-four wind facilities reported 2,039 bird fatalities spanning 17 orders, 41 families, 128 species, and 22 trophic guilds (for a complete list of reported bird species, see S1 Appendix). Excluding unidentified fatalities, passerines were the most common order (78%), wood warblers (Parulidae) were the most common family (28%; Table 2), and red-eyed vireos (*Vireo olivaceus*) were the most common species (18%; Table 3). Forty-nine percent of bird fatalities were lower canopy, foliage-gleaning insectivores. Four species of hawks, two vultures, two owls, and one falcon were reported totaling 96 fatalities (5%). One bald eagle (*Halieetus leucocephalus*) was reported. Two upland game birds, wild turkey (*Meleagris gallopavo*) and ruffed grouse (*Bonasa umbellus*), were reported totaling 75 fatalities (4%). Of the 128 species, 123 were protected under the Migratory Bird Treaty Act [62] and 103 were protected under the Neotropical Migratory Bird Conservation Act [63]. Migrants accounted for 59% of fatalities, partial migrants accounted for 34%, and residents accounted for 8%.

Forty-nine percent of all bird fatalities were composed of just eight species: red-eyed vireo, golden-crowned kinglet (*Regulus satrapa*), magnolia warbler (*Setophaga magnolia*), black-throated blue warbler (*Setophaga caerulescens*), ruffed grouse, yellow-rumped warbler (*Setophaga coronata*), common yellowthroat (*Geothlypis trichas*), and turkey vulture (*Cathartes aura*) (Table 3). Over half of all species were reported less than five times. Only three species were reported at more than 50% of facilities (red-eyed vireo, golden-crowned kinglet, magnolia warbler) and only 19 species were reported at 25% or more of facilities. The most widespread species was red-eyed vireo, reported at 89% of facilities. By proportion to each species' total US population, black-throated blue warblers were most heavily impacted with 25 fatalities per one million individuals. They were followed by sharp-shinned hawks (*Accipiter striatus*;

**Table 3. Eight most frequent species of bird mortality at wind turbines in the Northeastern US.**

| Species | Number of Fatalities | % of Total Mortality | Total Population[a] | Population Change 1970–2017[a] |
|---|---|---|---|---|
| Red-eyed Vireo | 376 | 21.72% | 130 million | +43% |
| Golden-crowned Kinglet | 122 | 7.05% | 130 million | -25% |
| Magnolia Warbler | 99 | 5.72% | 39 million | +51% |
| Black-throated Blue Warbler | 59 | 3.41% | 2.4 million | +163% |
| Ruffed Grouse | 55 | 3.18% | 18 million | +31% |
| Yellow-rumped Warbler | 51 | 2.95% | 150 million | 0% |
| Common Yellowthroat | 49 | 2.83% | 81 million | -34% |
| Turkey Vulture | 40 | 2.31% | 6.7 million | +186% |

[a]Rosenberg et al. [64]

**Table 4. Species of bat mortality at wind turbines in the Northeastern US.**

| Species | Fatalities |
|---|---|
| Big Brown Bat | 21 |
| Eastern Red Bat | 87 |
| Hoary Bat | 188 |
| Little Brown Bat | 16 |
| Silver-haired Bat | 99 |
| Unidentified Bat | 7 |
| **Grand Total** | **418** |

23/1,000,000) and black-billed cuckoos (*Coccyzus erthropthalmus*; 21/1,000,000). In contrast, only three red-eyed vireos per one million individuals were reported as fatalities.

Twenty-two wind facilities reported 418 bat fatalities, seven of which were unidentified. Five species were identified and three species of migratory tree bats, hoary (*Lasiurus cinereus*), silver-haired (*Lasionycteris noctivagans*), and eastern red (*Lasiurus borealis*), accounted for over 90% of bat fatalities (Table 4). These three were reported at more than 60% of facilities that reported bat mortality, while the remaining two bats, big brown (*Eptesicus fuscus*) and little brown (*Myotis lucifugus*), were reported at less than 25% of facilities that reported bat mortality.

Monthly mortality for birds appeared highest in September with a secondary peak in May, while monthly mortality for bats had a single peak in August (Fig 2). Monthly mortality also varied for some orders, migration types, and species (Fig 2). Three multiple mortality events with more than 20 fatalities in a single night were reported. Reports indicated that the largest event (80 fatalities) was associated with heavy fog. No bats were reported during any multiple mortality event.

## Spatial patterns

Our secondary dataset of bird and bat fatalities with observed fall distance included 1,999 records (birds = 984, bats = 1,015). We removed from our analyses nine bird fatality records with fall distances greater than 100 m, and 136 bird fatality records and 9 bat fatality records lacking species classification. For our construction of the birds-only models, we removed 16 records associated with non-migratory species. Additionally, we were unable to find supplementary information on migration timing in Birds of the World [65] for four bird species (each representing one record): least flycatcher (*Empidonax minimus*), purple finch (*Haemorhous purpureus*), redhead (*Aythya americana*), and ring-billed gull (*Larus delawarensis*).

Our global models showed a range of performance in terms of fit (Table 5). The birds-only global model's predictor variables accounted for 31% of the variation in fall distance and the bats-only global model's predictors explained only 3%. In each of our global models, the random effect of facility ID accounted for a substantial proportion of the variation explained, as evidenced by the disparity between conditional and marginal pseudo $R^2$ values. Among the all-taxa (S1 Table), birds-only (S2 Table), and bats-only candidate models, we found support for six, four, and three models, respectively (Tables 6–8).

Model averaging revealed varying support for the influence of turbine dimensions and animal characteristics on fall distance (Table 9). Turbine hub height, which we removed from the bats-only model for collinearity, did not prove influential in either all-taxa or birds-only model averages. However, when interacting with rotor diameter in the all-taxa dataset, hub height did appear to influence fall distance; as rotor diameter and turbine hub height increased, so did fall distance. Without interaction, rotor diameter also increased fall distance of birds-only and all-taxa. The magnitude and confidence intervals of these coefficients indicated greater

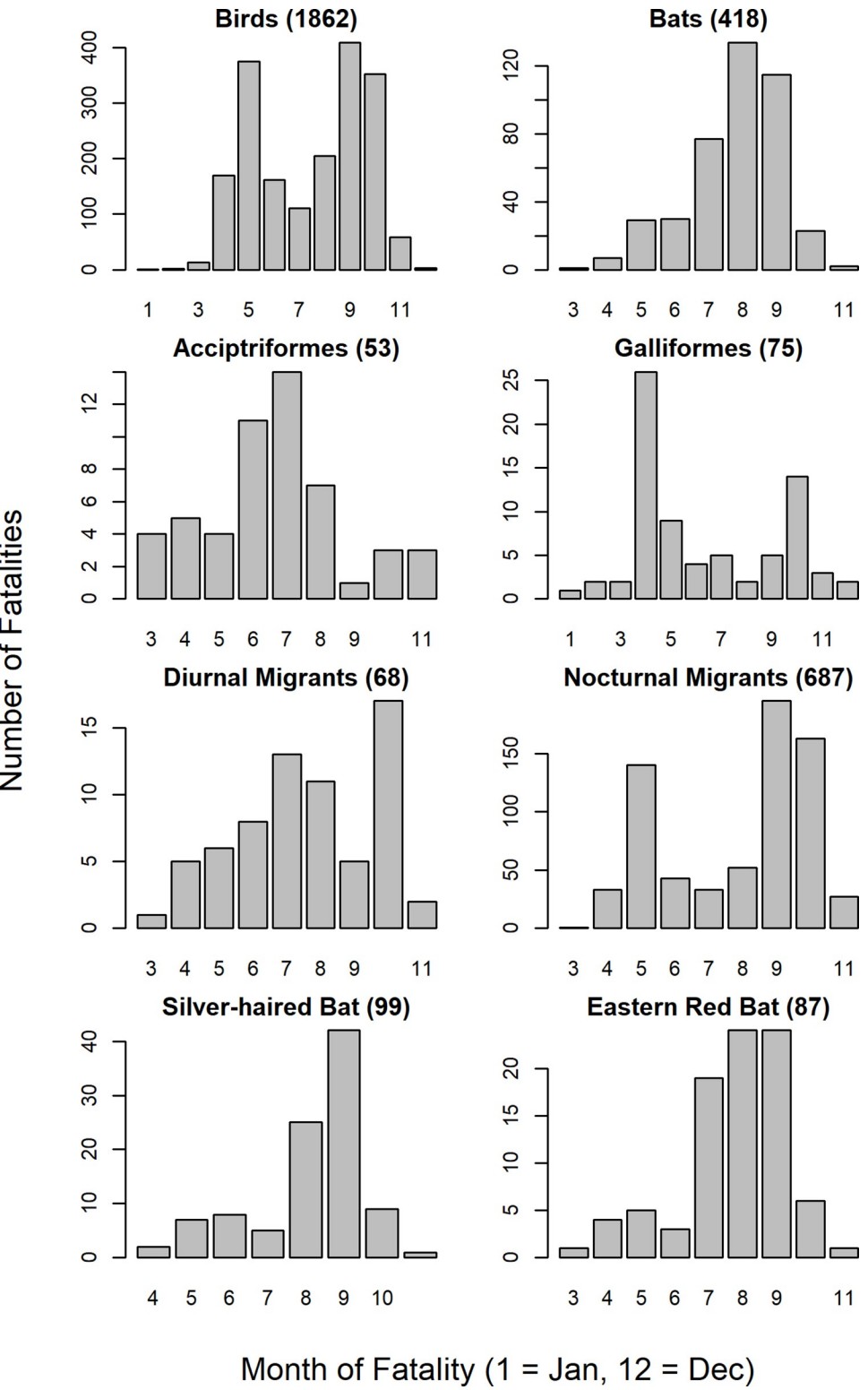

**Fig 2. Fatalities per month for birds and bats, Acciptriformes (hawks and eagles) and Galliformes (ruffed grouse and wild turkey), golden-crowned kinglets and black-throated blue warblers, and silver-haired bats and eastern red bats.** For monthly fatality histograms of the 10 most frequent families, see S1–S10 Figs.

**Table 5. For each global model, variables, random effect variance (RE Var) and standard deviation (RE SD) of the facility ID random effect, number of observations, number of wind facilities (n groups), dispersion parameter (φ), marginal pseudo R² (R²m), and conditional pseudo R² (R²c).**

| Model | Variables[a] | RE Var | RE SD | n obs | n groups | φ | R²m | R²c |
|---|---|---|---|---|---|---|---|---|
| all-taxa | A, D, H, M, D:H | 0.02 | 0.15 | 1845 | 14 | 8.32 | 0.13 | 0.44 |
| birds-only | D, H, M, MD, MT, D:H | 0.03 | 0.16 | 819 | 14 | 9.12 | 0.31 | 0.59 |
| bats-only | D, M, D:H | 0.05 | 0.22 | 1006 | 9 | 7.27 | 0.03 | 0.52 |

[a]Animal type (A), turbine rotor diameter (D), turbine hub height (H), mass (M), turbine diameter:height (D:H), migration distance (MD), migration timing (MT)

strength in these relationships in the birds-only model versus the all-taxa model. We also found strong support within the birds-only model-averaged coefficients for the influence of migration distance on fall distance. Short-distance migrants, those that traveled no more than 15 degrees in latitude [39], tended to fall farther from turbines following collision than long-distance migrants. Additionally, within the all-taxa model-averaged coefficients, we found that animal type (i.e., bird/bat) predicted fall distance, with birds associated with greater distances.

Species mass proved to be the one predictor variable influencing fall distance across all three sets of model-averaged coefficients. We found that fall distance increased with species body mass for birds-only, bats-only, and all-taxa. This result was most evident in birds-only and least evident in all-taxa. However, the bats-only and all-taxa mass coefficients featured relatively narrow confidence bands compared to the birds-only coefficient.

Examining the modality of fall distance distributions, we found bird (n = 975; p < 0.001) and bat (n = 1015; p < 0.001) density distributions were significantly multi-modal. The SiZer maps for both distributions suggested two peaks (i.e., bimodality). For bird fall distance, these modes were located at 2.37 m and 30.64 m, with an antimode (i.e., trough) at 8.80 m (Fig 3). Modes of bat fall distance showed less separation and were located at 17.00 m and 26.55 m, with the intervening antimode at 23.25 m (Fig 3).

## Discussion

### Taxonomic and temporal patterns

To our knowledge, our dataset represents the largest sample size of bird mortality in the Northeastern US. Generally, the species composition of our data confirm findings from

**Table 6. Variables, number of parameters (K), delta Quasi-AIC (ΔQAIC), log-likelihood (LL), and QAIC weights for all-taxa top model set.** Shaded rows represent models with uninformative parameters.

| Variables[a] | K | ΔQAIC | LL | $w_i$ |
|---|---|---|---|---|
| A, M | 4 | 0.00 | -12838.36 | 0.28 |
| A | 3 | 0.74 | -12849.75 | 0.19 |
| A, D, M | 5 | 1.45 | -12836.08 | 0.13 |
| A, H, M | 5 | 1.99 | -12838.33 | 0.10 |
| A, D | 4 | 2.21 | -12847.56 | 0.09 |
| A, H | 4 | 2.73 | -12849.73 | 0.07 |
| A, D, H, M | 6 | 3.41 | -12835.89 | 0.05 |
| A, D, H | 5 | 4.16 | -12847.36 | 0.03 |
| A, D, H, M, D:H | 7 | 4.81 | -12833.42 | 0.03 |
| A, D, H, D:H | 6 | 5.57 | -12844.88 | 0.02 |

[a]Animal type (A), turbine rotor diameter (D), turbine hub height (H), mass (M), turbine diameter:height (D:H), migration distance (MD), migration timing (MT)

**Table 7. Variables, number of parameters (K), delta Quasi-AIC (ΔQAIC), log-likelihood (LL), and QAIC weights (wᵢ) for birds-only top model set.** Shaded rows represent models with uninformative parameters.

| Variables[a] | K | ΔQAIC | LL | $w_i$ |
|---|---|---|---|---|
| M, MD | 4 | 0.00 | -6126.48 | 0.18 |
| MD | 3 | 0.49 | -6137.83 | 0.14 |
| D, M, MD | 5 | 1.31 | -6123.32 | 0.10 |
| D, MD | 4 | 1.79 | -6134.63 | 0.07 |
| M, MD, MT | 5 | 1.95 | -6126.25 | 0.07 |
| H, M, MD | 5 | 1.96 | -6126.30 | 0.07 |
| MD, MT | 4 | 2.14 | -6136.22 | 0.06 |
| H, MD | 4 | 2.46 | -6137.68 | 0.05 |
| D, M, MD, MT | 6 | 3.26 | -6123.09 | 0.04 |
| D, H, M, MD | 6 | 3.30 | -6123.28 | 0.04 |
| D, MD, MT | 5 | 3.43 | -6133.00 | 0.03 |
| D, H, MD | 5 | 3.77 | -6134.55 | 0.03 |
| H, M, MD, MT | 6 | 3.91 | -6126.06 | 0.03 |
| H, MD, MT | 5 | 4.10 | -6136.05 | 0.02 |
| D, H, M, MD, D:H | 7 | 4.80 | -6121.03 | 0.02 |
| D, H, MD, D:H | 6 | 5.13 | -6131.66 | 0.01 |
| D, H, M, MD, MT | 7 | 5.25 | -6123.05 | 0.01 |
| D, H, MD, MT | 6 | 5.42 | -6132.94 | 0.01 |

[a]Animal type (A), turbine rotor diameter (D), turbine hub height (H), mass (M), turbine diameter:height (D:H), migration distance (MD), migration timing (MT)

existing literature. Previously, the largest review of bird mortality at wind facilities was a technical report by the American Wind Wildlife Institute (AWWI) [34]. Though AWWI did not specifically summarize data for the Northeastern US, their North (including southeastern Canada) and East (including eastern Texas) regions include the extent of our data and serve as a useful comparison.

Similar to other studies, species in our data were primarily passerines, and nearly half of all fatalities comprised a small number of species (n ≤ 8) [17, 34]. Species were also comparable to other studies, predominantly red-eyed vireo, golden-crowned kinglet, and magnolia warbler. Raptors made up just 5% of fatalities, similar to other reports in the eastern US [34], but

**Table 8. Variables, number of parameters (K), delta Quasi-AIC (ΔQAIC), log-likelihood (LL), and QAIC weights (wᵢ) for bats-only top model set.** Because ΔQAIC was less than six for all models, the top model set is the same as the full model set. Shaded rows represent models with uninformative parameters.

| Variables[a] | K | ΔQAIC | LL | $w_i$ |
|---|---|---|---|---|
| M | 3 | 0.00 | -6300.13 | 0.44 |
| none | 2 | 1.61 | -6313.25 | 0.20 |
| D, M | 4 | 1.76 | -6299.27 | 0.18 |
| D | 3 | 3.34 | -6312.27 | 0.08 |
| D, M, D:H | 5 | 3.64 | -6298.82 | 0.07 |
| D, D:H | 4 | 5.22 | -6311.81 | 0.03 |

[a]Animal type (A), turbine rotor diameter (D), turbine hub height (H), mass (M), turbine diameter:height (D:H), migration distance (MD), migration timing (MT)

**Table 9. Model-averaging results for all-taxa, birds-only, and bats-only top model sets.** Included are standardized coefficient estimates (Estimate), unconditional standard errors (SE), and 95% confidence intervals (CI) for each model set. Shaded rows represent variables with confidence intervals crossing zero.

| Model | Variables[c] | Estimate | SE | CI | |
|---|---|---|---|---|---|
| | | | | Lower | Upper |
| All-Taxa | (Intercept) | 3.12 | 0.06 | 3.00 | 3.25 |
| | A[a] | 0.15 | 0.01 | 0.13 | 0.18 |
| | M | 0.04 | 0.01 | 0.02 | 0.06 |
| | D | 0.19 | 0.08 | 0.03 | 0.36 |
| | H | -0.04 | 0.04 | -0.11 | 0.03 |
| | D:H | 0.23 | 0.09 | 0.04 | 0.41 |
| Birds-Only | (Intercept) | 3.23 | 0.06 | 3.10 | 3.35 |
| | M | 0.24 | 0.05 | 0.14 | 0.34 |
| | MD[b] | 0.21 | 0.02 | 0.17 | 0.25 |
| | D | 0.30 | 0.11 | 0.09 | 0.51 |
| Bats-Only | (Intercept) | 3.05 | 0.09 | 2.86 | 3.23 |
| | M | 0.07 | 0.01 | 0.04 | 0.09 |
| | D | -0.12 | 0.08 | -0.29 | 0.05 |

[a]Reference category bird

[b]Reference category short-distance migrant

[c]Animal type (A), turbine rotor diameter (D), turbine hub height (H), mass (M), turbine diameter:height (D:H), migration distance (MD)

lower than raptor mortality along the west coast of the US [34, 66]. Our records included one bald eagle fatality, a species that has not been reported in the Northeastern US in other reviews [19, 34].

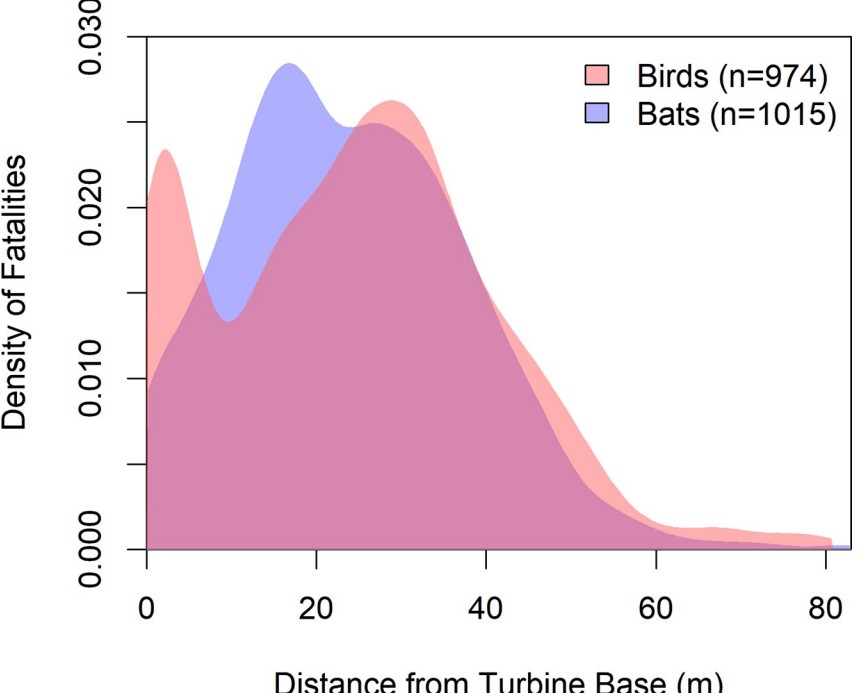

**Fig 3. Density distributions for the fall distance of birds and bats at wind turbines in the Northeastern US.** For density distributions of the 10 most frequent families, see S11–S20 Figs.

Relative to population size, black-throated blue warbler, sharp-shinned hawk, and black-billed cuckoo experienced the highest levels of mortality. At 25 or less fatalities per million individuals, even these are likely too low to significantly impact population trajectories, but this effect may intensify as the number of wind facilities increases. New world warblers, the most common family in our data, have declined by 38% over the past 50 years (Table 2), but of the species frequently observed in our data set, most populations are either stable or increasing (Table 3). Though we make these population comparisons using raw counts, in actuality, not all carcasses were recovered during carcass searchers. Detection probability is variable [31] but is typically higher than 0.25 and often higher than 0.50 for birds [40–42, 67], suggesting that population level impacts remain low.

Patterns of bat mortality in our data are generally consistent with those observed in previous studies. Three migratory tree bats (hoary bat, silver-haired bat, and eastern red bat) were somewhat more common (90% of all bat fatalities in our data) than reported elsewhere [22, 68], but our data represent a slightly different geographic extent. Additionally, our sample size is small, as relatively few facilities reported data for bats. Other bat species have been reported in the Northeastern US, but make up a low proportion of total fatalities [22]. We are unable to interpret the population-level impacts of our reported mortality because of the general lack of published bat demographic data [24], though previous research has identified wind energy mortality as a major potential driver of population declines in some species [23]. To understand the conservation implications of bat mortality at wind facilities fully, a broad effort must be made to expand and improve knowledge of bat species' population sizes and trends.

For both birds and bats, our data showed expected patterns of magnitude and annual timing of mortality. The peaks in bird mortality in September and May are generally consistent with other studies [17, 34], and presumably correspond to fall and spring migration activity [69]. Similarly, the single peak in bat mortality in August has been previously noted [22] and corresponds to migration timing of tree bats [70]. A global review reported that wind turbines cause multiple mortality events for bats (greater than 10 bats at a given locality within a few days [71]), but our data included none. This may be due to limited sample size, geographic differences, or biased reporting.

## Spatial patterns

Our analysis suggesting that fall distance increases with bird body mass represents a significant improvement on previous studies. To our knowledge, ours is the first attempt to analyze the relationship between bat body mass and fall distance. Though we acknowledge that our generalized body mass values are crude estimates for the actual mass of individual carcasses, our use of a continuous measure instead of a subjective categorical classification (e.g., large/small carcasses [27]) creates greater certainty that mass positively influences fall distance, and opens the possibility for easier comparison with future studies. To the extent that our findings on mass can be compared to past studies, they appear to corroborate similar findings from models [27]. Moreover, while AWWI's [34] distance distributions did not appear to differ between large and small birds, this may have been the result of overly broad size bins. A practical implication of these findings is that post-construction mortality studies concerned with large-bodied birds (e.g., raptors) should consider implementing wider search radii [27] or adjusting their weighting of area searched during analysis to reflect the increased potential of discovering larger species farther from turbines. Proper understanding of the relationship between species mass and fall distance can not only serve general understanding of wind turbine collisions, but also guide wind industry efforts towards compliance with government protection of particular species (e.g., eagle species in the US).

The positive association we found between bird fall distance and turbine rotor diameter and the additional finding that fall distance for birds and the all-taxa group increased with the interaction of increasing rotor diameter and hub height validate prior models and experiments suggesting that turbine height and blade length are positively related to fall distance [27, 33]. Given this general association, as the wind industry transitions to larger and more efficient turbine models, there may be need for wider search radii in post-construction mortality monitoring, or, alternatively, recalculation of typical fall distance distributions used to inform search efforts. Past studies have reported that bat mortality increases with hub height [72] but have been inconclusive concerning bird mortality, showing negative [16, 73], neutral [72], and positive [5] relationships between mortality and turbine height. Though collision mortality rates and fall distances are not necessarily directly related, increases in mortality rates at larger turbines may be harder to detect if carcasses are falling farther from the turbines as a result of increased turbine size.

Exploring facility-specific factors is also clearly critical to understanding rates and patterns of turbine collisions. In our models, the random effect of facility ID accounted for substantially more variation in fall distance than species or turbine traits. Because topography [74, 75], weather [76, 77]), and habitat [78, 79] influence bird flight, particularly during migration, we hypothesize that such local factors may affect how far birds and bats fall from the turbines with which they collide. Further, similar to collision rates, we expect patterns in fall distance may be driven by the interaction of species-specific factors (e.g., sensorial perception, behavior, etc.) and wind facility characteristics [18]. For example, spatial patterns may be different at wind facilities with turbines arranged in a line (e.g., along a mountain ridge) than at those with turbines more evenly spaced in a group (e.g., in agricultural fields). These facility-specific drivers may be a fruitful area for future research. A second possible explanation is facility-to-facility variation in survey technique and reporting culture, which, again, may be worthy of additional research. Variation between individual turbines within a wind facility may also affect spatial patterns of fall distance. For a few wind facilities that contained turbines of different sizes, we used only the most common turbine dimensions in our analysis, and this may have contributed to the variation accounted for by the random effect of facility ID. Despite this variation, managers can still expect findings such as the positive relationship between turbine size and fall distance to be generally true at specific facilities.

In all our final models, there was a notable proportion of variation in fall distance left unexplained. In the case of bats, we attribute this observation at least partly to the chaotic dynamics of falling carcasses [80]. However, we also speculate that a sizeable portion of this variation is the result of temporary and event-specific factors such as wind velocity and direction, direction of the blade sweep (i.e. upswing or downswing), blade velocity, and bird or bat flight velocity.

## Differences between birds and bats

Examining birds and bats separately can improve understanding of patterns in fall distances. As shown, birds tend to land farther from turbines than bats. This observation could theoretically be attributable to birds having a much higher range of masses, another influential variable in our models. However, in our all-taxa global model, we found that animal type and body mass had a low correlation (R = -0.09), which indicates the role of taxon-specific factors beyond size. Likewise, the multimodal distributions of fall distances for birds and bats argue in favor of different mechanisms. Though both distributions in our study were significantly multi-modal, the location and intensity of their peaks differed notably. Bats displayed two peaks, either side of 20 m, that were comparable in density and likely not biologically

significant. In contrast, birds exhibited a primary peak near 30 m and a lesser peak around 2.5 m. Similar secondary peaks in bird carcasses near the turbine base have been sparsely reported in other studies, but not rigorously analyzed or discussed [27, 34, 81].

We suggest this discrepancy in distance distribution between birds and bats is due to birds colliding with turbines poles, in addition to the turbines blades. It is well established that birds collide with stationary structures [82–84] and bird carcasses have been found beneath non-operational wind turbines [28, 85]. However, bats have rarely been found under towers and other structures [86, 87] and have not been found under non-operational turbines [68, 85]. Several investigations have shown that neither birds [18, 88, 89] nor bats [90, 91] are able to avoid swift-moving turbine blades by reacting to their presence at close range. Thus, in the absence of non-blade caused mortality, a similar distance distribution would be expected for both birds and bats at active wind turbines. Nonetheless, we found that 10% of all bird carcasses were located just 2 m away from the turbine base. Though most fatalities have been presumed to occur upon collision with moving turbine blades [29, 92], birds, especially nocturnal migrants, likely also struggle to detect the presence of turbine poles, whereas bats are assumedly able to avoid static turbine poles using echolocation.

It is also likely that some species, based simply on life history, primarily collide with turbine poles and rarely collide with blades themselves [6, 20, 93]. Carcasses of low-flying upland game birds, which are unlikely to ever fly into the rotor zone, have been routinely found during carcass searches [34, 94–97] and Stokke et al. [98] found substantial evidence that willow ptarmigan (*Lagopus lagopus*) frequently collide with turbine poles in Norway. Further, the distance distribution for willow ptarmigan mirrored ours, showing a high peak directly next to the turbine pole (Fig 3) [98]. However, though our dataset included 75 ruffed grouse and wild turkey, upland gamebirds represented only a minority of the near-turbine carcasses.

Small migratory songbirds comprised the majority of carcasses within 2 m of turbine polls, and may collide with poles at relatively high numbers. This has not been previously discussed, despite this group of birds being commonly reported colliding with other stationary structures, such as buildings [84] and communication towers [82, 83]. Our data suggested the phenomenon of turbine pole collisions was variable among taxonomic families of small migratory songbirds. For example, vireos show a pronounced peak in fall distance near turbines, new world sparrows and new world warblers only minor peaks, and kinglets and thrushes none at all (S12–S16 Figs). Because carcasses farther from turbines are less likely to be detected [30], this could lead to inaccurate mortality estimates of different bird groups. For instance, vireos, which showed a strongly right-skewed distance distribution, could be overestimated if this distribution was not incorporated into mortality estimates. We cannot presently explain these discrepancies, but think this matter is worthy of further investigation.

The concept of turbine pole collisions is not novel, as bird collisions with other tall structures (e.g., communication towers, buildings) are very common [2]. However, apart from brief discussion of collisions by upland gamebirds [6], pole collisions across avian taxa have received very minor attention by either researchers or governing agencies. Instead, an implicit assumption in studies examining wildlife mortality at wind facilities has been that mortality of both bird and bat species is primarily attributable to collision with moving turbine blades [5, 16, 17, 27]. However, our findings suggest that static turbines and towers also pose a risk to birds simply because they are tall structures that are difficult for birds to detect and avoid. As such, managers seeking to mitigate bird mortality at wind facilities should consider the application of mitigation technologies and existing best practices for the siting of other vertical structures (e.g., telecommunication and meteorological towers). Further, research and regulations regarding bird collisions with turbines, especially small passerines, may benefit from an expanded view of turbines as being very similar to other tall structures.

Inference from our results is limited as our data were both spatially and temporally variable, spanning nine years and multiple states. Additionally, the distance distribution we present is based on various mortality survey procedures, surveyor expertise, curtailment regimes (purposeful reduction in turbine operation and electricity generation), and turbine models and sizes. Year to year, many individual wind facilities varied between reporting data from incidental findings and mortality surveys, and some also varied the intensity of mortality surveys and their curtailment regimes. Because facilities voluntarily reported bat fatalities, fewer reported bat mortality as thoroughly or consistently as bird mortality. Additionally, because this dataset represents only 49% of Northeastern US turbines, there may be trends that were not observed. For example, almost no turbines along the coast of the Great Lakes and Massachusetts are included in this dataset, and trends may be different for shorebirds or seabirds in certain geographic areas and areas with different weather patterns.

Despite these limitations, our data represents the largest and most complete understanding of bird mortality caused by wind facilities in the Northeastern US. Many of the mortality reports submitted to the US Fish and Wildlife Service are not available elsewhere. Further, large-scale studies of turbine-related mortality are lacking in peer-reviewed literature, and large-scale summaries like we report here have been called for by wind-wildlife researchers [99]. Though wind facilities in our data did not use a standard search radius, all studies in our dataset searched to at least 20 m, so we are confident that, at a minimum, the distance distribution for both birds and bats within about 20 m, including the bimodal finding for birds, is an accurate representation of the true distribution.

## Implications

Our analysis of species composition supports the conclusion that collisions at wind facilities do not currently present a significant threat to populations of many bird species. Our analysis of fall distance provides important empirical evidence that bird fall distance increases with body mass and that both bird and bat fall distance increase with turbine size. As wind turbines increase in size, search plot radii and estimations of missed search areas will need to be adjusted to account for a higher number of carcasses farther away from turbines. Because some birds, especially small passerines, may collide regularly with turbine poles, it is possible that some species-specific estimates of mortality may be overestimated. Finally, as wind energy expands globally, we propose that policy-makers, managers, and researchers will benefit from not only recognizing the collision risk of moving turbine blades, but also in viewing wind turbines as presenting a threat similar to other tall structures.

## Supporting information

**S1 Appendix. Bird species and total fatalities from 44 wind facilities in the Northeastern US.** Data from reports submitted to the US Fish and Wildlife Service via either a special purpose utility or scientific collecting permit between 2008 and 2017.
(DOCX)

**S1 Fig. Monthly timing of fatalities of new world warblers (Parulidae) at wind turbines.** Data from reports submitted to the US Fish and Wildlife Service by 44 wind facilities in the Northeastern US. Sample size given in parentheses.
(PNG)

**S2 Fig. Monthly timing of fatalities of vireos (Vireonidae) at wind turbines.** Data from reports submitted to the US Fish and Wildlife Service by 44 wind facilities in the Northeastern

US. Sample size given in parentheses.
(PNG)

**S3 Fig. Monthly timing of fatalities of kinglets (Regulidae) at wind turbines.** Data from reports submitted to the US Fish and Wildlife Service by 44 wind facilities in the Northeastern US. Sample size given in parentheses.
(PNG)

**S4 Fig. Monthly timing of fatalities of grouse and allies (Phasianidae) at wind turbines.** Data from reports submitted to the US Fish and Wildlife Service by 44 wind facilities in the Northeastern US. Sample size given in parentheses.
(PNG)

**S5 Fig. Monthly timing of fatalities of new world sparrows (Passerellidae) at wind turbines.** Data from reports submitted to the US Fish and Wildlife Service by 44 wind facilities in the Northeastern US. Sample size given in parentheses.
(PNG)

**S6 Fig. Monthly timing of fatalities of thrushes (Turdidae) at wind turbines.** Data from reports submitted to the US Fish and Wildlife Service by 44 wind facilities in the Northeastern US. Sample size given in parentheses.
(PNG)

**S7 Fig. Monthly timing of fatalities of hawks and eagles (Accipitridae) at wind turbines.** Data from reports submitted to the US Fish and Wildlife Service by 44 wind facilities in the Northeastern US. Sample size given in parentheses.
(PNG)

**S8 Fig. Monthly timing of fatalities of new world vultures (Cathartidae) at wind turbines.** Data from reports submitted to the US Fish and Wildlife Service by 44 wind facilities in the Northeastern US. Sample size given in parentheses.
(PNG)

**S9 Fig. Monthly timing of fatalities of cuckoos (Cuculidae) at wind turbines.** Data from reports submitted to the US Fish and Wildlife Service by 44 wind facilities in the Northeastern US. Sample size given in parentheses.
(PNG)

**S10 Fig. Monthly timing of fatalities of flycatchers (Tyrannidae) at wind turbines.** Data from reports submitted to the US Fish and Wildlife Service by 44 wind facilities in the Northeastern US. Sample size given in parentheses.
(PNG)

**S11 Fig. Density plot of distance from turbine of fatalities of new world warblers (Parulidae) at wind turbines.** Data from publicly available reports and from reports submitted to the US Fish and Wildlife Service by 44 wind facilities in the Northeastern US. Sample size given in parentheses.
(TIFF)

**S12 Fig. Density plot of distance from turbine of fatalities of vireos (Vireonidae) at wind turbines.** Data from publicly available reports and from reports submitted to the US Fish and Wildlife Service by 44 wind facilities in the Northeastern US. Sample size given in parentheses.
(TIFF)

**S13 Fig. Density plot of distance from turbine of fatalities of kinglets (Regulidae) at wind turbines.** Data from publicly available reports and from reports submitted to the US Fish and Wildlife Service by 44 wind facilities in the Northeastern US. Sample size given in parentheses. (TIFF)

**S14 Fig. Density plot of distance from turbine of fatalities of thrushes (Turdidae) at wind turbines.** Data from publicly available reports and from reports submitted to the US Fish and Wildlife Service by 44 wind facilities in the Northeastern US. Sample size given in parentheses. (TIFF)

**S15 Fig. Density plot of distance from turbine of fatalities of new world sparrows (Passerellidae) at wind turbines.** Data from publicly available reports and from reports submitted to the US Fish and Wildlife Service by 44 wind facilities in the Northeastern US. Sample size given in parentheses. (TIFF)

**S16 Fig. Density plot of distance from turbine of fatalities of hawks and eagles (Accipitridae) at wind turbines.** Data from publicly available reports and from reports submitted to the US Fish and Wildlife Service by 44 wind facilities in the Northeastern US. Sample size given in parentheses. (TIFF)

**S17 Fig. Density plot of distance from turbine of fatalities of new world vultures (Cathartidae) at wind turbines.** Data from publicly available reports and from reports submitted to the US Fish and Wildlife Service by 44 wind facilities in the Northeastern US. Sample size given in parentheses. (TIFF)

**S18 Fig. Density plot of distance from turbine of fatalities of cuckoos (Cuculidae) at wind turbines.** Data from publicly available reports and from reports submitted to the US Fish and Wildlife Service by 44 wind facilities in the Northeastern US. Sample size given in parentheses. (TIFF)

**S19 Fig. Density plot of distance from turbine of fatalities of grouse and allies (Phasianidae) at wind turbines.** Data from publicly available reports and from reports submitted to the US Fish and Wildlife Service by 44 wind facilities in the Northeastern US. Sample size given in parentheses. (TIFF)

**S20 Fig. Density plot of distance from turbine of fatalities of flycatchers (Tyrannidae) at wind turbines.** Data from publicly available reports and from reports submitted to the US Fish and Wildlife Service by 44 wind facilities in the Northeastern US. Sample size given in parentheses. (TIFF)

**S1 Dataset. Data used in modeling fall distance of bird and bat mortality.** (CSV)

**S2 Dataset. Data used for taxonomic composition and temporal patterns.** (XLSX)

**S1 Table. Variables, number of parameters, delta Quasi-AIC ($\Delta$QAIC), QAIC weights ($w_i$), and log-likelihood (LL) for all-taxa full model set.** (DOCX)

**S2 Table. Variables, number of parameters, delta Quasi-AIC (ΔQAIC), QAIC weights ($w_i$), and log-likelihood (LL) for birds-taxa full model set.**
(DOCX)

**S1 Code. R code for data analysis.**
(R)

## Acknowledgments

This paper was written by DC as part of the Directorate Resource Assistant Fellows Program (DFP) with the US Fish and Wildlife Service. Pamela Toschik provided valuable support and leadership in development of this DFP project. Meghan Sadlowski generously assisted with data collation and other logistical matters. The findings and conclusions in this article are those of the authors and do not necessarily represent the views of the US Fish and Wildlife Service or the US Department of Agriculture.

## Author Contributions

**Conceptualization:** Daniel Y. Choi, Thomas W. Wittig, Bryan M. Kluever.

**Data curation:** Daniel Y. Choi.

**Formal analysis:** Daniel Y. Choi, Thomas W. Wittig.

**Funding acquisition:** Thomas W. Wittig, Bryan M. Kluever.

**Investigation:** Daniel Y. Choi, Thomas W. Wittig.

**Methodology:** Daniel Y. Choi, Thomas W. Wittig.

**Project administration:** Daniel Y. Choi, Thomas W. Wittig, Bryan M. Kluever.

**Resources:** Daniel Y. Choi, Thomas W. Wittig.

**Software:** Daniel Y. Choi, Thomas W. Wittig.

**Supervision:** Daniel Y. Choi, Thomas W. Wittig, Bryan M. Kluever.

**Validation:** Daniel Y. Choi, Thomas W. Wittig.

**Visualization:** Daniel Y. Choi.

**Writing – original draft:** Daniel Y. Choi, Thomas W. Wittig, Bryan M. Kluever.

**Writing – review & editing:** Daniel Y. Choi, Thomas W. Wittig, Bryan M. Kluever.

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
