## [Decision Letter · Decision Letter 0]

22 Apr 2020

PONE-D-20-07129

An evaluation of bird and bat mortality at wind turbines in the Northeastern United States

PLOS ONE

Dear Mr. Choi,

Thank you for submitting your manuscript to PLOS ONE. After careful consideration, we feel that it has merit but does not fully meet PLOS ONE’s publication criteria as it currently stands. Therefore, we invite you to submit a revised version of the manuscript that addresses the points raised during the review process. All three reviewers have raised important issues about the methodology, such as the staitistical approach, which need to be revised.

There are other structural and grammatical issues that have been pointed out, that would improve significantly the work.

The reviewers see the merit in the research and the importance of the topic being analysed, so we encourage the authors to resubmit, and we would appreciate receiving your revised manuscript by Jun 06 2020 11:59PM. To enhance the reproducibility of your results, we recommend that if applicable you deposit your laboratory protocols in protocols.io, where a protocol can be assigned its own identifier (DOI) such that it can be cited independently in the future. For instructions see: http://journals.plos.org/plosone/s/submission-guidelines#loc-laboratory-protocols

We look forward to receiving your revised manuscript.

Kind regards,

Vanesa Magar, Ph.D.

Academic Editor

PLOS ONE

Journal Requirements:

Reviewers' comments:

Reviewer's Responses to Questions

**Comments to the Author**

1. Is the manuscript technically sound, and do the data support the conclusions?

Reviewer #1: Partly

Reviewer #2: Yes

Reviewer #3: Partly

2. Has the statistical analysis been performed appropriately and rigorously? 

Reviewer #1: Yes

Reviewer #2: Yes

Reviewer #3: No

3. Have the authors made all data underlying the findings in their manuscript fully available?

Reviewer #1: Yes

Reviewer #2: Yes

Reviewer #3: Yes

4. Is the manuscript presented in an intelligible fashion and written in standard English?

Reviewer #1: No

Reviewer #2: Yes

Reviewer #3: Yes

5. Review Comments to the Author

Reviewer #1: Choi et al. present data on taxonomic composition and fall distances of bird and bat carcasses from wind-energy facilities in the northeastern United States. I have three overall comments, which are fleshed out further in my specific comments below. First, I found the manuscript generally hard to read. There was much extraneous text not directly related to the topic of this study. Second, I found objective 1 (species composition) to be the weakest part of this manuscript because this is a topic that has already been well studied and the results the authors show are already well known (which is consistent with the authors repeatedly saying that their “results confirm prior studies”). It is sometimes appropriate to repeat and validate prior studies, but the authors didn’t provide a justification as to why they thought this was necessary to do. Furthermore, it isn’t clear to me how much of the data they present on the species composition of carcasses has already been published in prior studies. Finally, in regards to objectives 2-3 (on fall distances): these results may be greater interest to readers since there hasn’t been as much study of this topic. However, although this topic has been less studied, the authors didn’t convince me why it is important to understand. I also found their discussion of how wildlife managers could use such results to be vague and hand wavy.

Specific comments

Lines 17-19: It would be helpful if the authors explained why an understanding of the spatial patterns of fatalities would be useful. What problem would such data help to solve? How would wildlife managers, etc. use such data?

Lines 30-31: I’m not sure it’s fair to say that it is thought that turbines are a threat only because of their motion. Certainly, it is well known that other stationary structures can cause bird fatalities.

Lines 31-32: Is this a finding of the present manuscript or prior studies? It sounds like the authors are referring to prior studies since they don’t present data in the abstract related to this already well-documented topic. However, it doesn’t make sense to have the conclusion of the abstract related to prior studies. The conclusion should be related to the present study.

Lines 33-34: This sentence is vague and much too general. This abstract doesn’t seem to interpret the results that this study presents. Instead of this sentence, I suggest the authors provide an interpretation or discuss the implications of the results of their results regarding fall distance, body mass, turbine height, etc.

Line 40: A good recent citation to add here and elsewhere in this manuscript is Allison et al. 2019. Impacts to wildlife of wind energy siting and operation in the United States. Issues in Ecology. Report number 21.

Line 59: Is this citation correct? Perhaps add additional citations here to support the sentence on lines 57-59.

Lines 59-62: This is probably true for most bird species, but might not be true for some such as raptors (Allison et al. 2019; Katzner et al. 2020).

Line 63: This sentence is too general. Wind-energy fatalities of some bird species are of concern, and some species of bats aren’t thought to be impacted by wind turbines.

Lines 64-67: I’m not sure I agree with the logic of these sentences. Bats are more active at low than high wind speeds, but has that been shown to be the cause of slightly higher bat than bird mortality in the eastern US as these sentences imply? Is there a reference the authors could cite to support this claim?

Lines 69-71: True, but the authors should acknowledge (since some readers won’t know) that the tree-roosting bats (i.e., eastern red, hoary, and silver-haired bats) that experience the greatest numbers of fatalities at wind-energy facilities don’t seem to be affected by WNS.

Lines 71-72: I’m not sure what this sentence means. Certainly bats are killed at wind-energy facilities in regions of the U.S. where WNS occurs. Please clarify.

Lines 76-79: True, but this sentence is tagged on to the end of this paragraph and it’s relevance to this paragraph and study is unclear.

Lines 86-87: This is the topic of this paper, but it wasn’t mentioned until the 5th paragraph of the Introduction. Most readers won’t have the patience to wade through 5th paragraphs of background material that is only indirectly related to the topic of this manuscript.

Lines 92-95: This (i.e. “dynamics and nuances”) is much too vague. The authors should be clear and specific about why/how information on the spatial arrangement of carcasses would benefit monitoring, mitigation, and conservation.

Lines 97-98: Objective 1 is something that we arguably know a lot about already (e.g. Allison et al. 2019, AWWI reports that are cited, many other peer-reviewed publications and consultant reports). If the authors truly feel that this is a topic that hasn't been adequately address then their Introduction should articulate that what their study will address/accomplish that hasn’t been addressed/accomplished in prior studies.

Lines 98-99: How is fall distance defined and why is it important if it varies between bats and birds? How might the authors expect it to vary? As with their objective 1, the Introduction hasn’t set up this objective.

Lines 99-100: Why do the authors expect these factors might influence spatial arrangement of carcasses?

Line 110: What is meant by “migration status”?

Lines 130-131: What is the justification for censoring/removing records from non-migratory species?

Lines 202-203: I know what the authors mean, but fog isn’t a direct cause of fatalities.

Lines 232-237: What do the error bars in panel A indicate? What do the small tick marks in panel B and the thicker black bars in panel C indicate? Also, what do lower vs. higher cHeight values indicate (lower values = shorter towers?)?

Lines 265-267: Although statistically significant, I wonder if the difference in the two modes of bat fall distance are biologically significant.

Line 273: Is the fact that the results of this study vary little from existing literature because many of the same data were used in this study as in prior studies? It would be helpful if somewhere in the manuscript (probably the methods) the authors addressed if/how the datasets used in the present manuscript differ from previously published datasets.

Lines 317-320: I’m not sure that this is true. What evidence supports the statement that use of generalized body mass values creates greater uncertainty than categorical classifications. I imagine there can be a great deal of variation in a factor such as body mass.

Lines 320-322: This sentence implies that post-construction mortality studies often focus on finding carcasses of a particular species, but I’m not sure that’s the case. They are typically interested in finding all bird or bat (or both) carcasses, as I understand.

Lines 322-325: Figure 5 in the 2019 AWWI report shows similar fall distances for small and large birds. Why might the results of the present study differ from that report?

Lines 331-333: Please specify how this finding would be relevant to post-construction studies.

Lines 339-340: Correct. Facility ID is very important, which implies that it will be difficult to make management recommendations (as the authors do earlier in the text) without an understanding of what it is about different facilities (or survey techniques, etc., among facilities) that influences fall distance.

Lines 368-392: These are interesting findings.

Lines 393-402: We know that mortality at wind-energy facilities is relatively minor relative to other sources of mortality (cats, buildings) for most species of birds, so I’m not sure that managers who want to reduce bird fatalities would really pay much attention to whether fatalities at wind-energy facilities are caused by the pole or blades.

Lines 393-395: Do these studies really make this assumption? It is well known that birds are killed by other tall structures, so I find it odd that these studies would assume that bird mortality at wind-energy facilities is primarily attributable to collision with blades.

Lines 403-414: This paragraph doesn’t seem very related to the results presented in this manuscript.

Lines 427-428: How many? It feels like this is a point the authors should have made in their Introduction.

Lines 437-451: None of these implications are derived from the results presented in this manuscript.

Reviewer #2: The authors present a paper that looks at bat and bird fatalities at wind turbine facilities and runs a series of GLMMs to explore whether turbine specifications and species attributes such as mass or migratory status impact the distance at which the carcasses are found from the base of the turbine. Overall I believe this is worthy of publication and is a useful exploration that could help managers make regulatory decisions on how to mitigate bird and bat deaths at wind farms. I have several comments on the writing and some of how the analysis is presented, detailed below.

Line 36: This paragraph is about how wind facilities impact wildlife, so it seems awkward to begin by mentioning how much wind power is growing. Instead, it would make more sense to begin the paragraph with something like "Wind turbine facilities can have direct and indirect negative impacts on wildlife.... then, after you've summarized the literature on different species responses to wildlife, you can end the paragraph with the fact that turbines are increasing across the world, tripling in the last decade, and therefore are expected to have further impacts on wildlife.

I think the second and third paragraph should either be combined, or the two paragraphs should be separated by birds and bats, respectively. Jumping back and forth especially in the second paragraph is confusing for me as a reader.

Line 66: I think rotor-swept zone should be defined here, it is not common knowledge what this is. Additionally, please define what speed "lower wind speed" is - what is the average wind speed considered low?

Line 69: white nose syndrome has been around for around 15 years now, I am not sure calling it 'recent' is accurate. Perhaps you could say something like "Since 2006, the fatal fungal pathogen white-nose syndrome has caused massive population declines in once-common bat species in the northeastern and midwestern US. Additional mortalities from wind turbine collisons could impact recovery of these species.."

Line 71: Please explain why that is. I am assuming it is because WNS deaths are so high, it is difficult to tease wind turbine deaths apart from that, right?

Line 72: an additional note: hoary bats are not particularly impacted by WNS as it mostly kills cave hibernating bats, of which hoary bats are not. But a reader not savvy on bats may see the sentence beforehand and this one and say, wait a minute - didn't you just say we couldn't tease apart wind turbine deaths from WNS deaths? This discrepancy should be cleared up in the paragraph.

Lines 80-85: This is a 2-sentence paragraph and I am unsure what the exact purpose of the paragraph is. I think if it remains in the paper it needs to be fleshed out a little more, maybe describe additional mitigation efforts that are not as effective?

Line 95: I think it would be helpful here to describe exactly why investigating the spatial arrangement of fatalities will benefit mitigation/conservation. "dynamics and nuances" is a little vague. Can you mention some specific things it could address?

Line 96: not super important but ~2000 data points is not considered large by some.. consider deleting - you could also call it long-term as it is from 2008-2017

Line 100: spatial arrangement is still a sort of vague term to me.. is it fall distance? if so, perhaps that can go in parentheses after 'spatial arrangement'

Lines 102-109: This entire paragraph is written in passive tense which I find to be a bit awkward. Additionally, instead of listing out each state I think it would be much better to have a map highlighting the states where data came from and if possible, areas where wind turbines are located within the state. I recognize the latter may not be possible due to privacy concerns and endangered species location reporting, but a map at least showing the states would be beneficial especially for those readers not familiar with the USA.

Line 110: This paragraph needs a topic sentence. Perhaps something like, "We supplemented the fatality data records with information that would serve as predictor variables or ways to subset data for our models"

Line 122: I'm confused why a Poisson distribution was used with fall distance rounded when fall distance could have been used in a Gaussian GLMM just as easily? Is it because you wanted to avoid predicting negative values at all? If so this should be explained here.

Line 157: I guess I am confused as to what was done with the estimated location of these modes? I understand that you identified them, but I'd like some detail as to why identifying multi-modality is important for the analysis, what it could do to inference, and what was done to address it.

Line 192: Vespertillionidae are the only bats that exist in the regions you have data from

Line 211: Just a preference, but censored is a little awkward. Consider changing it to "removed"

Figures 3&4: I'd really like to have the response curves be shown on back-transformed variables so that it is easier to interpret these figures... right now I have no idea what the average bird or bat mass was, so these figures don't say much to me. If I saw the back-transformed masses, I could easily see the relationship and interpret it without trying to think about what the average mass is.

Reviewer #3: This is an excellently written manuscript on factors associated with wind turbines that cause mortality in both birds and bats. I really enjoyed reading this manuscript and the editing is one of the best that I have encountered over my many years as a peer reviewer. So, kudos to the authors on such excellent prose. The only big issue I had with the manuscript was the flawed statistical approach. Other than that, this manuscript will make an excellent contribution to the literature once the statistical approach is revised.

Here are three general comments on the methods followed by my line edits.

1. How is fall distance defined? Is this the perpendicular distance from the center of the hub to the ground or does this include both the vertical and horizontal distances from the hub? The authors skim over this like it is common knowledge. Likewise, they include the term “nacelle” in Table 1 but do not mention that term, nor define it in the Methods.

2. The authors used wind facility as a random effect but how does this account for the number of turbines at the facility. Is turbine not nested within site? In particular, by including turbine ID as a random effect, the authors would not have to worry about the bias associated with assigning the facility with the most common hub height and rotor diameter.

3. The stepwise method of reducing model size has been discounted by many authors. Here are two examples of papers that show why this approach is longer acceptable. An information-theoretic approach using AIC would be much more appropriate in this context. As a result, I am unable to assess the Results section because of this flawed approach, unfortunately.

Mundry, R. & Nunn, C.L. (2009) Stepwise model fitting and statistical inference: turning noise into signal pollution. Am Nat, 173, 119-123.

Whittingham, M.J., Stephens, P.A., Bradbury, R.B. & Freckleton, R.P. (2006) Why do we still use stepwise modelling in ecology and behaviour? Journal of Animal Ecology, 75, 1182-1189.

Line Edits

15 The authors should be more specific about who these benefits apply to. Presumably to humans but not to wildlife.

24 Unnecessary to include “Generalized linear mixed models revealed that”. Simply begin with “Turbine size…”

53, Remove “For bats,” since it is redundant with the rest of the sentence.

55-56 Inconsistent “to” vs “and” when demarcating a range of values. Use one style.

59-62 I think there are many scientists that would disagree with this statement that minimizes the effects of turbines. While it might not have an effect at the scale of overall bird mortality, there are likely key species where it does have a large effect, partly because of site location. The overall effects are likely to rise with increasing densities of wind farms at continental scales.

130 It’s unclear what the censoring of non-migratory species means. Are the authors simply reporting that they did not include non-migratory species in the analysis? Censor typically means one removed datapoints from a particular end of a distribution, hence my confusion.

148 The authors should cite Ameijeiras-Alonso et al. 2019 after the reference to ACR.

Ameijeiras-Alonso, J., Crujeiras, R.M. & Rodríguez-Casal, A. 2019. Mode testing, critical bandwidth and excess mass. TEST 28, 900–919.

149 It’s unclear to me how distances can be 0 if this represents the fall distance.

150 Insert “to” so it reads “due to the right-skewness”.

151 Not clear what “meaningful calculation of modes beyond this value” means. Are the authors implying that they limited the maximum distance to 60 m? Earlier (line 127), however, the authors wrote that they limited fall distances to 100 m. Please clarify both the terminology and the distance issues.

152 Likewise, I don’t understand the statement “We ran all other function parameters at default values”. Is this something inherent in package multimode? If so, provide more explanation about what these default values are and how they are used.

162 passerines were already introduced at line 52. Thus, the parenthetical (Passeriformes) should appear at first mention, not here.

168 Remove “identified” since it is redundant.

169 I think the authors should use different terminology here to avoid confusion. Normally, “federally listed” means an endangered or threatened species. In this case, they simply mean that the species is among those covered by the MBTA or NMBC.

171 Again, “identified” is unnecessary to include. You’ve already said at line 162 that you are excluding those that were not identified. And you did not provide such caveats in other parts of this paragraph when referring to percentage fatalities (insectivores, upland birds, etc).

176 And, again, no need for “identified”. If it were not identified, it could not be assigned to species.

193 Scientific name for hoary was already given at line 73. As before, “excluding unidentified” is implied when one is reporting by species. Please delete.

197 Is this 25% of facilities that reported bat mortality? If so, please clarify.

198 The ordering of Table 4 is not intuitive. The first and last rows are redundant. All bats are in Chiroptera so this seems unnecessary to list. Why not just list each species, followed by a row with Unidentified Bat? I suggest deleting “CHIROPTERA, Unknown, and Vespertilionidae”. This information adds nothing.

211 As before, are the authors using “censored” in this context as a synonym for “removed”? And, just to clarify, were any birds with distances >100 m set to 100 m or 60 m or deleted? Ambiguous as worded.

213 The authors use the word “removed” in this context and this is much clearer to the reader.

232 Bats should be plural in the figure caption. Please indicate what the error bars in panel A represent.

241 Please indicate what the error bars in panel A represent.

243 Typo in top-performing.

286 Too low to significantly impact total populations, given the current density and spatial extent of existing turbines. If the number of facilities increases, this effect may increase.

289 Please provide a citation for the statement that “most populations are either stable or increasing”.

298 Need to add a hyphen to the compound modifier “population-level impacts”.

306 Please provide a citation for the Sep and May peaks in spring and fall migration activity.

309 Just to clarify, does “multiple mortality” mean that two individuals of the same species were found under the same turbine? Or at the same facility? Unclear terminology.

6. PLOS authors have the option to publish the peer review history of their article (what does this mean?). If published, this will include your full peer review and any attached files.

Reviewer #1: No

Reviewer #2: No

Reviewer #3: No

---

## [Author Response · Author response to Decision Letter 0]

5 Jun 2020

Dear Dr. Magar,

My coauthors and I are grateful for the feedback provided by the reviewers, which we feel has greatly strengthened our manuscript. Enclosed is our revised manuscript titled “An evaluation of bird and bat mortality at wind turbines in the Northeastern United States” (PONE-D-20-07129) that we wish to have considered for resubmission as an original research article in PLOS ONE. We believe that we have addressed the comments and concerns put forth by the reviewers. For example, we have revised the Introduction section, removing information not immediately relevant and adding two paragraphs which provide clear justification for our objectives. In both our Introduction and Discussion sections, we have made it more evident why results from our study are relevant to managers and researchers. Finally, we have adjusted our statistical methodology in accordance with reviewer suggestions. Below, we respond to each reviewer’s comments individually. We have kept the original line numbers associated with each comment but in our response have included line numbers that reference the updated manuscript.

This resubmission represents original work that is not under consideration for publication elsewhere. Data in this manuscript are not included in previous publications nor will they be submitted for publication elsewhere. We are willing to cover page chargers and any other costs associated with the publication of the manuscript. This paper aims to make a contribution to the fields of conservation science, ornithology, mammalogy, and animal behavior, by reporting on patterns associated with bird and bat mortality at wind facilities, with an emphasis on the spatial arrangement of carcasses. To our knowledge, our dataset represents the largest ever compiled for bird and bat mortality at wind facilities in the Northeastern United States, an area where wind energy production is rapidly increasing. In addition, our empirical support for bird collisions with turbines poles in addition to blades is novel and germane for scientists, managers, and those involved in the wind power industry. 

Sincerely, 

Daniel Choi

No comments from the academic editor.

Reviewer #1 General Comments

General Comment 1

Choi et al. present data on taxonomic composition and fall distances of bird and bat carcasses from wind-energy facilities in the northeastern United States. I have three overall comments, which are fleshed out further in my specific comments below. First, I found the manuscript generally hard to read. There was much extraneous text not directly related to the topic of this study. 

Response: Thank you for pointing out that we have included information not specifically relevant to this study. We have removed extraneous text from the introduction and discussion. For example, we removed four lines from the third paragraph (lines 59-74) of our introduction, which focused on bats. Additionally, we removed paragraphs in our introduction and discussion which mentioned mitigation efforts. We also completely rewrote our implications section so that it clearly reflects results from the present study.

General Comment 2

Second, I found objective 1 (species composition) to be the weakest part of this manuscript because this is a topic that has already been well studied and the results the authors show are already well known (which is consistent with the authors repeatedly saying that their “results confirm prior studies”). It is sometimes appropriate to repeat and validate prior studies, but the authors didn’t provide a justification as to why they thought this was necessary to do. Furthermore, it isn’t clear to me how much of the data they present on the species composition of carcasses has already been published in prior studies.

Response: Thank you for pointing this weakness out. We have revised our introduction to provide justification for examining species composition (objective 1) and have clarified how our data differs from what is currently published. Prior to this study, the two most significant reviews of mortality, and thereby species composition, were those by Erickson et al. in 2016 [17] and the American Wind Wildlife Institute in 2019 [34]. These studies summarized mortality by region. For example, their eastern North America region extended from eastern Texas to New York. Within this region, Erickson et al. and AWWI included data from eight and 20 wind facilities. This is a relatively small number considering the geographic extent of this region. Additionally, each of these studies specifically called for improved reviews to corroborate their findings. For example, AWWI specifically referred to their results as “tentative.” Therefore, we are able to offer substantially improved picture of species composition in the Northeastern US where we look at data from 44 wind facilities. Due to confidentiality of reports, we cannot be sure how many reports in our dataset overlap with that of AWWI. However, because we are able to include so many more facilities than AWWI, we are confident that at least most of our data is from facilities not included in their report.

General Comment 3

Finally, in regards to objectives 2-3 (on fall distances): these results may be greater interest to readers since there hasn’t been as much study of this topic. However, although this topic has been less studied, the authors didn’t convince me why it is important to understand. I also found their discussion of how wildlife managers could use such results to be vague and hand wavy.

Response: Thank you for making us aware of these points regarding support for objectives 2 and 3 and our discussion. We revised our introduction and discussion and believe it is now much stronger. First, we have combined objectives 2 and 3 into a single objective which is now to “examine whether animal type (bird/bat), body size, migration behavior, or turbine size influence the fall distances of carcasses associated with turbines” (lines 109-110). We have added a paragraph (lines 75-93) to our introduction that explains why understanding the patterns and sources of variation in fall distance are important to designing effective search protocols and accurately estimating mortality. We have also revised our discussion to reflect this reasoning. For example, we point out that for species such as vireos, which have sharp peaks in their distance distribution near the turbine base, mortality may potentially by overestimated and that estimates must account for skewed distance distributions (lines 449-452).

Reviewer #1 Specific Comments

Lines 17-19: It would be helpful if the authors explained why an understanding of the spatial patterns of fatalities would be useful. What problem would such data help to solve? How would wildlife managers, etc. use such data?

Response: We have added a sentence to our abstract explaining why this is important: “Understanding how far carcasses typically fall from turbines is important in designing effective search protocols and in estimating total mortality” (lines 19-20). Additionally, we have added a paragraph (lines 75-93) to our introduction that explains this in detail.

Lines 30-31: I’m not sure it’s fair to say that it is thought that turbines are a threat only because of their motion. Certainly, it is well known that other stationary structures can cause bird fatalities.

Response: Thank you for pointing this out. Certainly, this is well known and it is intuitive that turbine poles are a source of mortality. Yet, to our knowledge, this has not been discussed in the literature. There are a few sparse mentions in reports of gamebirds which may collide with poles. However, this has not been discussed for other bird groups. Many papers and reports explicitly refer to only turbine blades. We have adjusted our abstract and discussion to better communicate this point. For example, our discussion paragraph on lines 453-465 begins with “The concept of turbine pole collisions is not novel, as bird collisions with other tall structures (e.g., communication towers, buildings) are very common [2]. However, apart from brief discussion of collisions by upland gamebirds [6], pole collisions across avian taxa have received very minor attention by either researchers or management agencies” (lines 453-456).

Lines 31-32: Is this a finding of the present manuscript or prior studies? It sounds like the authors are referring to prior studies since they don’t present data in the abstract related to this already well-documented topic. However, it doesn’t make sense to have the conclusion of the abstract related to prior studies. The conclusion should be related to the present study.

Lines 33-34: This sentence is vague and much too general. This abstract doesn’t seem to interpret the results that this study presents. Instead of this sentence, I suggest the authors provide an interpretation or discuss the implications of the results of their results regarding fall distance, body mass, turbine height, etc.

Response: Thank you for pointing out that the last two sentences (both lines 31-32 and 33-34) of our abstract do not reflect the conclusions of the present study and thank you for providing suggestions. We have altered the end of our abstract so that it ends with: “Bird and bat fall distance distributions were notably multimodal, but only birds exhibited a high peak near turbine bases, a novel finding we attribute to collisions with turbine poles in addition to blades. This phenomenon varied across bird species, with potential implications for the accuracy of mortality estimates. Although pole collisions for birds is intuitive, this phenomenon has not been formally recognized. This finding may warrant an updated view of turbines as a collision threat to birds because they are a tall structure, and not strictly as a function of their motion” (lines 29-35). Additionally, we included in earlier sentence regarding turbine size and fall distance: “This suggests that as turbines increase in size, a greater search radius will be necessary to detect carcasses” (lines 28-29).

Line 40: A good recent citation to add here and elsewhere in this manuscript is Allison et al. 2019. Impacts to wildlife of wind energy siting and operation in the United States. Issues in Ecology. Report number 21.

Response: Thank you for this valuable source. We have included this paper as a reference here and elsewhere throughout the manuscript.

Line 59: Is this citation correct? Perhaps add additional citations here to support the sentence on lines 57-59.

Response: Thank you for catching this incorrect citation. We have modified this sentence to refer to raptors only and we have added additional references: “There is particular concern regarding raptors of low abundance [2, 18, 19], whose low reproductive rates and high adult survival may hamper recovery from mortality at wind facilities [6, 20]” (lines 53-55).

Lines 59-62: This is probably true for most bird species, but might not be true for some such as raptors (Allison et al. 2019; Katzner et al. 2020).

Response: Thank you for this comment. Certainly, some species may be severely impacted by collision with wind turbines. We have modified the beginning of this sentence: “However, for most bird species, wind turbines contribute a small amount to total anthropogenic mortality [21] and may not significantly impact vital rates (e.g., annual survival probability) and state variables (e.g., abundance, density, occupancy) at the population level” (lines 55-58).

Line 63: This sentence is too general. Wind-energy fatalities of some bird species are of concern, and some species of bats aren’t thought to be impacted by wind turbines.

Response: We have modified this sentence to specifically refer to differences between passerine birds and migratory tree bats; “Compared to many passerine birds, wind turbines may have a more pronounced negative impact on migratory tree bats” (lines 61-62).

Lines 64-67: I’m not sure I agree with the logic of these sentences. Bats are more active at low than high wind speeds, but has that been shown to be the cause of slightly higher bat than bird mortality in the eastern US as these sentences imply? Is there a reference the authors could cite to support this claim?

Response: Thank you for pointing out the error in this argument. We have removed the second sentence which links increased bat mortality to lower wind speeds in the east.

Lines 69-71: True, but the authors should acknowledge (since some readers won’t know) that the tree-roosting bats (i.e., eastern red, hoary, and silver-haired bats) that experience the greatest numbers of fatalities at wind-energy facilities don’t seem to be affected by WNS.

Response: Thank you for this suggestion. We have added the following sentence: “However, it should be noted that migratory tree bats that experience the most mortality at wind facilities are not as affected by white-nose syndrome [26]” (lines 70-71).

Lines 71-72: I’m not sure what this sentence means. Certainly bats are killed at wind-energy facilities in regions of the U.S. where WNS occurs. Please clarify.

Response: In an effort to reduce the introduction following recommendation by Reviewer 1, this sentence has been deleted.

Lines 76-79: True, but this sentence is tagged on to the end of this paragraph and its relevance to this paragraph and study is unclear.

Response: This sentence has been removed.

Lines 86-87: This is the topic of this paper, but it wasn’t mentioned until the 5th paragraph of the Introduction. Most readers won’t have the patience to wade through 5th paragraphs of background material that is only indirectly related to the topic of this manuscript.

Response: Thank you for comment regarding the length and detail of the introduction. We attempted to write our manuscript in a standard general-to-specific format. We have reduced the first three paragraphs and removed the fourth. This is now paragraph four. Several of our reductions were the result of incorporating Reviewer 2’s specific comments for this Section. It may be important to note that not only did Reviewer 3 not take issue with the writing style/flow of the manuscript, they stated “This is an excellently written manuscript on factors associated with wind turbines that cause mortality in both birds and bats. I really enjoyed reading this manuscript and the editing is one of the best that I have encountered over my many years as a peer reviewer. So, kudos to the authors on such excellent prose.”

Lines 92-95: This (i.e. “dynamics and nuances”) is much too vague. The authors should be clear and specific about why/how information on the spatial arrangement of carcasses would benefit monitoring, mitigation, and conservation.

Response: We have greatly expounded this paragraph (lines 75-93), removed vague language, and included clear and specific sentences regarding the importance of studying spatial arrangement of carcasses. Specifically, we have focused our discussion on fall distance and have explained how this is importance in designing carcass search protocols and in accurately estimating mortality.

Lines 97-98: Objective 1 is something that we arguably know a lot about already (e.g. Allison et al. 2019, AWWI reports that are cited, many other peer-reviewed publications and consultant reports). If the authors truly feel that this is a topic that hasn't been adequately address then their Introduction should articulate that what their study will address/accomplish that hasn’t been addressed/accomplished in prior studies.

Response: Please see our response to an identical comment, Reviewer 1 General Comment 2.

Lines 98-99: How is fall distance defined and why is it important if it varies between bats and birds? How might the authors expect it to vary? As with their objective 1, the Introduction hasn’t set up this objective.

Response: Concerning objective 2, please see our response to an identical comment, Reviewer 1 General Comment 3. We have adjusted our introduction to answer these questions. Regarding the definition of fall distance, we have included a definition in the second sentence of the fourth paragraph of the introduction: “the horizontal distance between a carcass and the nearest turbine pole” (lines 76-77).

Lines 99-100: Why do the authors expect these factors might influence spatial arrangement of carcasses?

Response: Please see our response to an identical comment, Reviewer 1 General Comment 3. We have adjusted our introduction to answer these questions.

Line 110: What is meant by “migration status”?

Response: We have modified this sentence with parentheses following: “By species, we added migration status (i.e., resident, migratory, partial migrant)…” (line 128).

Lines 130-131: What is the justification for censoring/removing records from non-migratory species?

Response: Removing non-migratory birds was necessary to conduct likelihood ratio tests between models during our original stepwise model fitting. Had we left non-migratory birds in our analysis, those models with migratory-related variables would have had fewer data points (non-migratory species’ values being ‘NA’) than those without these variables, invalidating statistical comparison between models. Although we have changed our model fitting to an Information-Theoretic approach based off the recommendation of Review 3, missing data similarly confound model comparison in this approach. In revising our methodology, we have explained the model comparison justification for removing non-migratory birds (lines 136-151) and included a relevant citation (Nakagawa and Freckleton 2010 [47]).

Lines 202-203: I know what the authors mean, but fog isn’t a direct cause of fatalities.

Response: Thank you for pointing this out. We have revised this to indicate that fog was associated with mortality events, but did not specifically cause mortality: “Reports indicated that the largest event (80 fatalities) was associated with heavy fog” (lines 235-236).

Lines 232-237: What do the error bars in panel A indicate? What do the small tick marks in panel B and the thicker black bars in panel C indicate? Also, what do lower vs. higher cHeight values indicate (lower values = shorter towers?)?

Response: We have removed this figure. Following on the recommendation of Reviewer 3, we revised our methodology and fit our models according to an Information-Theoretic approach. The results showed several viable candidate models for the all-taxa, birds-only, and bats-only sets, leading us to perform model-averaging. Unfortunately, we were unable to find current software implementation for plotting model-averaged coefficients in a manner similar to our original response curves. However, we have added a table with model-averaged coefficients, standard errors, and 95% confidence interval, which we believe comparably conveys the direction and magnitude of the variable responses.

Lines 265-267: Although statistically significant, I wonder if the difference in the two modes of bat fall distance are biologically significant.

Response: Thank you pointing this out. We agree, and have included the following sentence in the first paragraph of our discussion section titled “Differences between Birds and Bats”: “Bats displayed two peaks, either side of 20 m, that were comparable in density and likely not biologically significant” (lines 417-418).

Line 273: Is the fact that the results of this study vary little from existing literature because many of the same data were used in this study as in prior studies? It would be helpful if somewhere in the manuscript (probably the methods) the authors addressed if/how the datasets used in the present manuscript differ from previously published datasets.

Response: Please see our response to an identical comment, Reviewer 1 General Comment 3. We have adjusted our introduction to answer these questions. Additionally, we have added the following to the first paragraph of our methods section: “We are unsure how many of these facilities may have also contributed data to studies such as those by AWWI [22, 34]. However, we include 44 total wind facilities, compared to 20 in the eastern region of North America included by AWWI [34]. Thus, the majority of wind facilities in our study contribute new data in examining species composition” (lines 117-121).

Lines 317-320: I’m not sure that this is true. What evidence supports the statement that use of generalized body mass values creates greater uncertainty than categorical classifications. I imagine there can be a great deal of variation in a factor such as body mass.

Response: Thank you for your comment. Undeniably, the mass of one individual is not the mass of all members of that species. This metric will vary within species based on sex, age, condition, etc. However, previous studies have used subjective categorizations of large and small, either based on mass and/or length [27, 34]. AWWI 2019 [34] divided large and small birds as having a total length of greater or less than 30 cm. We think our method of using average species masses improves greatly over just using two very broad categories. We have added a sentence mentioning this to the fourth paragraph of our introduction (lines 86-89). We have additionally modified the first paragraph of our discussion section titled “Spatial Patterns” to address this point (lines 361-377).

Lines 320-322: This sentence implies that post-construction mortality studies often focus on finding carcasses of a particular species, but I’m not sure that’s the case. They are typically interested in finding all bird or bat (or both) carcasses, as I understand.

Response: We have modified this sentence to suggest that particularly for large-bodied birds, search protocols must ensure they have a wide enough search radius: “A practical implication is that post-construction mortality studies concerned with large-bodied birds (e.g., raptors) should consider implementing wider search radii [27] or adjusting their weighting of area searched during analysis to reflect the increased potential of discovering larger species farther from turbines” (lines 370-374). Further, we have modified this paragraph to reiterate that our finding that mass is positively associated with fall distance corroborates findings from prior models.

Lines 322-325: Figure 5 in the 2019 AWWI report shows similar fall distances for small and large birds. Why might the results of the present study differ from that report?

Response: Thank you for drawing our attention to this figure. We have added a sentence to this paragraph (first paragraph of Discussion: Spatial Patterns) discussing this question: “Moreover, while AWWI’s [34] distance distributions for large and small birds did not appear to differ, this may have been the result of overly broad size bins” (lines 369-370).

Lines 331-333: Please specify how this finding would be relevant to post-construction studies.

Response: We have removed this sentence and have suggested that both “The positive association we found between bird fall distance and turbine rotor diameter and the additional finding that fall distance for birds and the all-taxa group increased with the interaction of increasing rotor diameter and hub height validate prior models and experiments suggesting that turbine height and blade length [27, 33] are positively related to fall distance” (lines 378-381).

Lines 339-340: Correct. Facility ID is very important, which implies that it will be difficult to make management recommendations (as the authors do earlier in the text) without an understanding of what it is about different facilities (or survey techniques, etc., among facilities) that influences fall distance.

Response: We have modified the ending of this paragraph (third paragraph of Discussion: Spatial Patterns) with: “Despite this variation, managers can still expect findings such as the positive relationship between turbine size and fall distance to be generally true at specific facilities” (lines 401-402). We agree that facility to facility variation prohibits very specific management recommendations. However, our general findings (e.g., fall distance positively influenced by mass, turbine size, etc.) can still be helpful in providing general predictive power, regardless of the variation at specific facilities.

Lines 368-392: These are interesting findings.

Response: Thank you for your interest.

Lines 393-402: We know that mortality at wind-energy facilities is relatively minor relative to other sources of mortality (cats, buildings) for most species of birds, so I’m not sure that managers who want to reduce bird fatalities would really pay much attention to whether fatalities at wind-energy facilities are caused by the pole or blades.

Response: We agree that managers may not pay specific attention to whether individual fatalities are caused by collisions with poles or blades. Because of this very fact, in combination with evidence for frequent pole collisions, we suggest that simply that managers may not need to focus specifically on the threat of moving turbine blades for many species. Instead, they should begin to view turbines as more of a threat simply because they are large structures like other towers. We have modified this paragraph (lines 453-465) accordingly. Even so, we think that managers’ interest in mitigating any collisions is evident in the large body of guidance developed by state and federal agencies, and in the intense efforts of wind projects themselves to understand this issue. Any understanding that could help carve away potential collisions is of interest to wind energy companies. 

Lines 393-395: Do these studies really make this assumption? It is well known that birds are killed by other tall structures, so I find it odd that these studies would assume that bird mortality at wind-energy facilities is primarily attributable to collision with blades.

Response: Please see our response to an identical comment, Reviewer 1 Specific Comment Lines 30-31.

Lines 403-414: This paragraph doesn’t seem very related to the results presented in this manuscript.

Response: Thank you for pointing this out. This paragraph has been removed.

Lines 427-428: How many? It feels like this is a point the authors should have made in their Introduction.

Response: We have added comments on this matter to the introduction (lines 94-106) and the methods (lines 112-121). We also discuss this in our response to Reviewer 1 General Comment 2.

Lines 437-451: None of these implications are derived from the results presented in this manuscript.

Response: Thank you for pointing this out. We have completed rewritten our implications section (lines 487-497) ensuring that they are clearly derived from the results in this manuscript.

Reviewer #2 General Comments

General Comment 1

The authors present a paper that looks at bat and bird fatalities at wind turbine facilities and runs a series of GLMMs to explore whether turbine specifications and species attributes such as mass or migratory status impact the distance at which the carcasses are found from the base of the turbine. Overall I believe this is worthy of publication and is a useful exploration that could help managers make regulatory decisions on how to mitigate bird and bat deaths at wind farms. I have several comments on the writing and some of how the analysis is presented, detailed below.

Response: Thank you for your support of this paper. We appreciate your comments regarding the writing and analysis and respond to them individually below.

Reviewer #2 Specific Comments

Line 36: This paragraph is about how wind facilities impact wildlife, so it seems awkward to begin by mentioning how much wind power is growing. Instead, it would make more sense to begin the paragraph with something like "Wind turbine facilities can have direct and indirect negative impacts on wildlife.... then, after you've summarized the literature on different species responses to wildlife, you can end the paragraph with the fact that turbines are increasing across the world, tripling in the last decade, and therefore are expected to have further impacts on wildlife. I think the second and third paragraph should either be combined, or the two paragraphs should be separated by birds and bats, respectively. Jumping back and forth especially in the second paragraph is confusing for me as a reader.

Response: Thank you for your advice. We have followed your suggestion and moved the first sentence to the end of the paragraph (lines 46-47). We have additionally separated the second (lines 48-58) and third (59-74) paragraphs by birds and bats.

Line 66: I think rotor-swept zone should be defined here, it is not common knowledge what this is. Additionally, please define what speed "lower wind speed" is - what is the average wind speed considered low?

Response: Because we do not have sources for the argument presented here, we have removed this sentence, including both references to “rotor-swept zone” and “lower wind speeds.”

Line 69: white nose syndrome has been around for around 15 years now, I am not sure calling it 'recent' is accurate. Perhaps you could say something like "Since 2006, the fatal fungal pathogen white-nose syndrome has caused massive population declines in once-common bat species in the northeastern and midwestern US. Additional mortalities from wind turbine collisons could impact recovery of these species.."

Response: Thank you for your suggestion. We have modified this sentence to: “Further, since 2006, many bat species have been severely impacted by white-nose syndrome and additional mortality from wind facilities could impact recovery of species of special concern [25]” (lines 68-70).

Line 71: Please explain why that is. I am assuming it is because WNS deaths are so high, it is difficult to tease wind turbine deaths apart from that, right?

Response: In an effort to reduce the introduction as suggested by Reviewer 1 we have removed this sentence.

Line 72: an additional note: hoary bats are not particularly impacted by WNS as it mostly kills cave hibernating bats, of which hoary bats are not. But a reader not savvy on bats may see the sentence beforehand and this one and say, wait a minute - didn't you just say we couldn't tease apart wind turbine deaths from WNS deaths? This discrepancy should be cleared up in the paragraph.

Response: We have added the following sentence to address this: “However, it should be noted that migratory tree bats which experience the most mortality at wind facilities are not as affected by white-nose syndrome [26]” (lines 70-71).

Lines 80-85: This is a 2-sentence paragraph and I am unsure what the exact purpose of the paragraph is. I think if it remains in the paper it needs to be fleshed out a little more, maybe describe additional mitigation efforts that are not as effective?

Response: This paragraph has been removed.

Line 95: I think it would be helpful here to describe exactly why investigating the spatial arrangement of fatalities will benefit mitigation/conservation. "dynamics and nuances" is a little vague. Can you mention some specific things it could address?

Response: We have greatly clarified this statement. This was also mentioned by another reviewer and we have added an entire paragraph (lines 75-93) to address this.

Line 96: not super important but ~2000 data points is not considered large by some.. consider deleting – you could also call it long-term as it is from 2008-2017

Response: We have removed the adjective “large,” and the sentence now reads: “Here, we investigated a dataset of bird and bat fatalities at wind facilities located in the Northeastern US” (lines 107-108).

Line 100: spatial arrangement is still a sort of vague term to me.. is it fall distance? if so, perhaps that can go in parentheses after 'spatial arrangement'

Response: We have removed the term “spatial arrangement” and simply referred to “fall distances” instead (line 110).

Lines 102-109: This entire paragraph is written in passive tense which I find to be a bit awkward. Additionally, instead of listing out each state I think it would be much better to have a map highlighting the states where data came from and if possible, areas where wind turbines are located within the state. I recognize the latter may not be possible due to privacy concerns and endangered species location reporting, but a map at least showing the states would be beneficial especially for those readers not familiar with the USA.

Response: We have changed the tense of the paragraph (lines 112-121) to active voice, except for the first sentence (lines 112-113). But because the authors had little or no role in receiving fatality data, we stated that “the USFWS received records” in order to explicitly communicate that these records were submitted to the agency as a whole, and not to individuals. Additionally, thank you for your recommendation to include a map. We have included a map (Fig. 1; lines 122-126) showing the Northeast within the United States, the amount of fatalities reported from each state, and the locations of wind facilities present in the Northeast. We were not able to include locations of wind facilities in this study due to privacy concerns. 

Line 110: This paragraph needs a topic sentence. Perhaps something like, "We supplemented the fatality data records with information that would serve as predictor variables or ways to subset data for our models"

Response: We have added a topic sentence and altered the second sentence. The first two sentences now read: “We supplemented fatality records with classification information and model covariates. By species, we added migration status (i.e., resident, migrant, partial migrant) and population data [35], taxonomic classification information [36, 37], trophic guild information [38], timing (nocturnal/diurnal) and distance (long/short) of migration [39], and body mass [37-39]” (lines 127-130).

Line 122: I'm confused why a Poisson distribution was used with fall distance rounded when fall distance could have been used in a Gaussian GLMM just as easily? Is it because you wanted to avoid predicting negative values at all? If so this should be explained here.

Response: As shown in the density figures for birds and bats, fall distance approximates a Poisson distribution. Using a Gaussian link would have required the added and unnecessary step of transforming the fall distance data and may have led to poorer model performance (O’Hara & Kotze 2010). 

O’hara RB, Kotze DJ. Do not log‐transform count data. Methods in ecology and Evolution. 2010 Jun 1;1(2):118-22.

Line 157: I guess I am confused as to what was done with the estimated location of these modes? I understand that you identified them, but I'd like some detail as to why identifying multi-modality is important for the analysis, what it could do to inference, and what was done to address it.

Response: The identified modes allowed us to stronger inference that rotor blades and towers are two distinct sources of mortality at wind facilities for birds. Because of the novel nature of this finding, we felt it important not rely on visual observation of the distributions alone. It is also our hope that by offering this methodology, we may encourage future comparison of our results against other datasets and study areas

Line 192: Vespertillionidae are the only bats that exist in the regions you have data from

Response: We have removed reference to Vespertillionidae and the sentence now reads: “Five species were identified and three species of migratory tree bats, hoary, silver-haired (Lasionnycteris noctivagans), and eastern red (Lasiurus borealis), accounted for over 90% of bat fatalities (Table 4)” (lines 226-228).

Line 211: Just a preference, but censored is a little awkward. Consider changing it to "removed"

Response: Another reviewer also mentioned this and throughout the paper, “censored” has been replaced with “removed.”

Figures 3&4: I'd really like to have the response curves be shown on back-transformed variables so that it is easier to interpret these figures... right now I have no idea what the average bird or bat mass was, so these figures don't say much to me. If I saw the back-transformed masses, I could easily see the relationship and interpret it without trying to think about what the average mass is.

Response: We have removed these figures. Please see our response to a similar comment regarding figure interpretation, Reviewer 1 Specific Comment Lines 232-237.

Reviewer #3 General Comments

This is an excellently written manuscript on factors associated with wind turbines that cause mortality in both birds and bats. I really enjoyed reading this manuscript and the editing is one of the best that I have encountered over my many years as a peer reviewer. So, kudos to the authors on such excellent prose. The only big issue I had with the manuscript was the flawed statistical approach. Other than that, this manuscript will make an excellent contribution to the literature once the statistical approach is revised. Here are three general comments on the methods followed by my line edits.

Response: We are very grateful for your praise for this paper. We have addressed your comments regarding our statistical approach in our responses below.

General Comment 1

How is fall distance defined? Is this the perpendicular distance from the center of the hub to the ground or does this include both the vertical and horizontal distances from the hub? The authors skim over this like it is common knowledge. Likewise, they include the term “nacelle” in Table 1 but do not mention that term, nor define it in the Methods.

Response: We have added the following definition of fall distance to the fourth paragraph of our introduction: “the horizontal distance between a carcass and the nearest turbine pole” (line 77). We have removed mention of the term “nacelle.”

General Comment 2

The authors used wind facility as a random effect but how does this account for the number of turbines at the facility. Is turbine not nested within site? In particular, by including turbine ID as a random effect, the authors would not have to worry about the bias associated with assigning the facility with the most common hub height and rotor diameter.

Response: We readily agree that nesting turbine ID as a random effect within facility ID would likely improve model performance. However, it is an unfortunate reality of data reporting that many carcass records with fall distance information do not also include associated turbine ID. It is primarily for this reason that we have not attempted such nesting of random effects. Regarding the influence of the number of turbines at facilities on fall distance, we believe that the random effect for facility ID may still address this variation. The total number of facility turbines is a factor that, like general landscape and habitat, may vary between facilities, but not within facilities. Therefore, if the number turbines at facilities does have relevance to fall distance, it is captured in the facility ID random effect. 

General Comment 3

The stepwise method of reducing model size has been discounted by many authors. Here are two examples of papers that show why this approach is longer acceptable. An information-theoretic approach using AIC would be much more appropriate in this context. As a result, I am unable to assess the Results section because of this flawed approach, unfortunately.

Mundry, R. & Nunn, C.L. (2009) Stepwise model fitting and statistical inference: turning noise into signal pollution. Am Nat, 173, 119-123.

Whittingham, M.J., Stephens, P.A., Bradbury, R.B. & Freckleton, R.P. (2006) Why do we still use stepwise modelling in ecology and behaviour? Journal of Animal Ecology, 75, 1182-1189.

Response: Thank you for offering this critique. After consulting the referenced literature for background, we have fit our models according to an Information-Theoretic approach, and provided citations for our decisions on selection information criterion, delta threshold, and model averaging (lines 157-174).

Reviewer #3 Specific Comments

Line 15: The authors should be more specific about who these benefits apply to. Presumably to humans but not to wildlife.

Response: We have updated the sentence to specifically mention humans: “Wind energy offers substantial benefits to humans, but wind facilities can negatively affect wildlife, including birds and bats” (lines 15-16).

Line 24: Unnecessary to include “Generalized linear mixed models revealed that”. Simply begin with “Turbine size…”

Response: We have removed “Generalized linear mixed models revealed that” from the sentence (lines 27-28).

Line 53: Remove “For bats,” since it is redundant with the rest of the sentence.

Response: We have removed “For bats” from the sentence (lines 59-60).

Lines 55-56: Inconsistent “to” vs “and” when demarcating a range of values. Use one style.

Response: We have changed “between 140,000 to 679,000” changed to “between 140,000 and 679,000” (line 52).

Lines 59-62: I think there are many scientists that would disagree with this statement that minimizes the effects of turbines. While it might not have an effect at the scale of overall bird mortality, there are likely key species where it does have a large effect, partly because of site location. The overall effects are likely to rise with increasing densities of wind farms at continental scales.

Response: We agree that though populations of most bird species may not be affected significantly by collisions, some certainly are. We have modified the sentence to: “However, for most bird species, wind turbines contribute a small amount to total anthropogenic mortality [21] and may not be high enough to impact vital rates (e.g., annual survival probability) and state variables (e.g., abundance, density, occupancy) at the population level” (lines 55-58).

Line 130: It’s unclear what the censoring of non-migratory species means. Are the authors simply reporting that they did not include non-migratory species in the analysis? Censor typically means one removed datapoints from a particular end of a distribution, hence my confusion.

Response: Another reviewer also mentioned this. We meant to communicate that we removed data from the analysis and throughout the paper “censored” is replaced with “removed.”

Line 148: The authors should cite Ameijeiras-Alonso et al. 2019 after the reference to ACR.

Ameijeiras-Alonso, J., Crujeiras, R.M. & Rodríguez-Casal, A. 2019. Mode testing, critical bandwidth and excess mass. TEST 28, 900–919.

Response: We have moved the citation from the end of sentence to directly after mention of “ACR” (line 177).

Line 149: It’s unclear to me how distances can be 0 if this represents the fall distance.

Response: Distances of 0 represent carcasses that were found directly next to a turbine base. We have modified the sentence: “We set the lower limits for these tests at 0 m (carcasses with a distance of 0 m were found directly next to the turbine base) to reflect the non-negative nature of distance values” (lines 179-181)

Line 150: Insert “to” so it reads “due to the right-skewness”.

Response: We have revised this paragraph (lines 175-191), and this clause is no longer mentioned.

Line 151: Not clear what “meaningful calculation of modes beyond this value” means. Are the authors implying that they limited the maximum distance to 60 m? Earlier (line 127), however, the authors wrote that they limited fall distances to 100 m. Please clarify both the terminology and the distance issues.

Response: There was insufficient information (i.e. mortality records) beyond this distance for the ACR method to meaningfully calculate modes. We found that over 60 m, the algorithm began honing in on and identifying small clusters of observations as modes, when common sense suggested there were too little data to generalize these results. We are not implying that carcasses beyond 60 m are not meaningfully tied to turbine collision, but merely attempting to provide an honest accounting of the limitations of this analysis technique. We have revised our methods to expand our explanation for limiting mode testing to below 60 m (lines 175-191).

Line 152: Likewise, I don’t understand the statement “We ran all other function parameters at default values”. Is this something inherent in package multimode? If so, provide more explanation about what these default values are and how they are used.

Response: We would direct readers to our added supplementary R code for further detail on function use. We did not specify or explain default values in the manuscript body as doing so would have been redundant to referenced R package documentation, required many additional lines of text, and would not have advanced understanding or interpretation of the results.

Line 162: passerines were already introduced at line 52. Thus, the parenthetical (Passeriformes) should appear at first mention, not here.

Response: We have removed the references to Passeriformes and included it at the first mention of passerines.

Line 168: Remove “identified” since it is redundant.

Response: We have removed “identified.”

Line 169: I think the authors should use different terminology here to avoid confusion. Normally, “federally listed” means an endangered or threatened species. In this case, they simply mean that the species is among those covered by the MBTA or NMBC.

Response: We have changed the language to use “protected” instead of “federally listed.” The sentence now reads: “Of the 128 species, 123 were protected under the Migratory Bird Treaty Act [62] and 103 were protected under the Neotropical Migratory Bird Conservation Act [63]” (lines 202-204).

Line 171: Again, “identified” is unnecessary to include. You’ve already said at line 162 that you are excluding those that were not identified. And you did not provide such caveats in other parts of this paragraph when referring to percentage fatalities (insectivores, upland birds, etc).

Response: We have removed “identified.”

Line 176: And, again, no need for “identified”. If it were not identified, it could not be assigned to species.

Response: We have removed “identified.”

Line 193: Scientific name for hoary was already given at line 73. As before, “excluding unidentified” is implied when one is reporting by species. Please delete.

Response: We have removed the scientific name for hoary bat and removed “excluding unidentified.”

Line 197: Is this 25% of facilities that reported bat mortality? If so, please clarify.

Response: That is correct. We have updated the sentence to reflect this: “…less than 25% of facilities that reported bat mortality” (line 230).

Line 198: The ordering of Table 4 is not intuitive. The first and last rows are redundant. All bats are in Chiroptera so this seems unnecessary to list. Why not just list each species, followed by a row with Unidentified Bat? I suggest deleting “CHIROPTERA, Unknown, and Vespertilionidae”. This information adds nothing.

Response: We have followed your suggestion and removed “CHIROPTERA, Unknown, and Vespertilionidae” and we have moved “Unidentified Bat” to the bottom (line 231).

Line 211: As before, are the authors using “censored” in this context as a synonym for “removed”? And, just to clarify, were any birds with distances >100 m set to 100 m or 60 m or deleted? Ambiguous as worded.

Response: We have replaced “censored” with “removed” throughout the paper. We removed these records from the analysis entirely. We modified the sentence to reflect this: “We removed from our analyses nine bird fatality records with fall distances greater than 100 m, and 136 bird fatality records and 9 bat fatality records lacking species classification” (lines 243-246).

Line 213: The authors use the word “removed” in this context and this is much clearer to the reader.

Response: We have replaced “censored” with “removed” throughout the paper.

Line 232: Bats should be plural in the figure caption. Please indicate what the error bars in panel A represent.

Response: We changed “bat” to “bats.”

Line 241: Please indicate what the error bars in panel A represent.

Response: We have removed these figures. Please see our response to a similar comment regarding figure interpretation, Reviewer 1 Specific Comment Lines 232-237.

Line 243: Typo in top-performing.

Response: We changed “top-preforming” to “top-performing”

Line 286: Too low to significantly impact total populations, given the current density and spatial extent of existing turbines. If the number of facilities increases, this effect may increase.

Response: Yes, this impact may change as the number of wind facilities increases. We have added the following sentence to the third paragraph of our discussion section titled “Taxonomic and Temporal Patterns”: “At 25 or less fatalities per million individuals, even these are likely too low to significantly impact total population trajectories, but this effect may intensify as the number wind facilities increases” (lines 332-334).

Line 289: Please provide a citation for the statement that “most populations are either stable or increasing”.

Response: This reference was included in the table which contains this data. We have included a reference to that table (3; lines 209-210) in this sentence.

Line 298: Need to add a hyphen to the compound modifier “population-level impacts”.

Response: We have added a hyphen.

Line 306: Please provide a citation for the Sep and May peaks in spring and fall migration activity.

Response: We have added a reference and additionally modified the sentence to indicate that these peaks “presumably” correspond to migration activity (line 354).

Line 309: Just to clarify, does “multiple mortality” mean that two individuals of the same species were found under the same turbine? Or at the same facility? Unclear terminology.

Response: For bats, we defined multiple mortality events as greater than 10 bats at a given locality within a few days [71]. We have added this definition to our discussion (lines 356-359).

---

## [Decision Letter · Decision Letter 1]

4 Jul 2020

PONE-D-20-07129R1

An evaluation of bird and bat mortality at wind turbines in the Northeastern United States

PLOS ONE

Dear Dr. Choi,

Thank you for submitting your revision of the manuscript to PLOS ONE.  We agree with the reviewers that it has been revised very thoroughly, however two reviewers suggest minor revisions,  and one reviewer suggests major revisions. Therefore, after careful consideration, we recommend mejor revisions based on the recommendations and the list of revisions suggested by the reviewers, who have been very generous with their comments. We invite you to submit a revised version of the manuscript that addresses the points raised during this round. 

We look forward to receiving your revised manuscript.

Kind regards,

Vanesa Magar, Ph.D.

Academic Editor

PLOS ONE

Reviewers' comments:

Reviewer's Responses to Questions

**Comments to the Author**

1. If the authors have adequately addressed your comments raised in a previous round of review and you feel that this manuscript is now acceptable for publication, you may indicate that here to bypass the “Comments to the Author” section, enter your conflict of interest statement in the “Confidential to Editor” section, and submit your "Accept" recommendation.

Reviewer #1: All comments have been addressed

Reviewer #2: (No Response)

Reviewer #3: (No Response)

2. Is the manuscript technically sound, and do the data support the conclusions?

Reviewer #1: Yes

Reviewer #2: Yes

Reviewer #3: Yes

3. Has the statistical analysis been performed appropriately and rigorously? 

Reviewer #1: Yes

Reviewer #2: Yes

Reviewer #3: No

4. Have the authors made all data underlying the findings in their manuscript fully available?

Reviewer #1: Yes

Reviewer #2: Yes

Reviewer #3: Yes

5. Is the manuscript presented in an intelligible fashion and written in standard English?

Reviewer #1: Yes

Reviewer #2: Yes

Reviewer #3: Yes

6. Review Comments to the Author

Reviewer #1: The authors have done a thorough job of addressing my prior comments and I believe the manuscript is now basically ready for publication. I have only a few minor specific suggestions:

Line 15: I suggest replacing “benefits to humans” with “environmental benefits” to help keep the manuscript focused and because I imagine there are some people who would disagree with this statement.

Line 59: I suggest replacing “Wind development” with something more clear and specific such as “Wind-energy development”.

Line 64: I suggest inserting “individuals” after “949,000”.

Reviewer #2: I have now read the revised manuscript by Choi et al. and find it to be MUCH improved from the first iteration. I appreciate the author's very careful consideration of all reviewers. The introduction reads very clearly and I believe the new analysis appraoch clears up the results and they are easier for the reader to understand.

Minor comment on Line 170: I think you mean < 6.

Reviewer #3: I thank the authors for submitting a much-improved manuscript. It is always encouraging when authors respond so diligently to reviewer suggestions / comments. In particular, I am glad to see that the stepwise analysis has been removed. That was obviously a deal breaker for me. I am satisfied with all of their responses to my edits / comments / queries. I do have some issues related to the revised manuscript, as follows:

Statistical Issues

1. L162 I believe Harrison’s method of looking at dispersion parameter > 1 only pertains to GLMs. The authors are using GLMMs so I would expect this to be the ratio of the sum of the squared Pearson residuals to the residual degrees of freedom, as outlined in Harrison (2014).

2. L167 Many information theorists will cringe at the “all combinations” approach of constructing candidate models. Dochtermann and Jenkins provide some good commentary on why one should not dredge candidate models. Dochtermann, N. & Jenkins, S. (2011) Developing multiple hypotheses in behavioral ecology. Behavioral Ecology and Sociobiology, 65, 37-45.

3. How confident are the authors in using package multimode? Has this package been vetted at all? The citation they give (reference 58) is to a preprint repository.

4. Table 6 looks to me like it includes uninformative parameters. For example, the top model AM and the 4th model AHM have identical log likelihoods. In this case, H is likely an uninformative parameter. You can see this again when you compare models A vs AH (they differ by one parameter but the log likelihoods are identical). Again, H is an uninformative parameter. See also AD vs ADH. Please refer to Figure 1 in Leroux, S.J. (2019) On the prevalence of uninformative parameters in statistical models applying model selection in applied ecology. PLoS ONE, 14, e0206711.

Please check all of your AIC tables for such uninformative parameters. I also see some in Table 7, and maybe in Table 8 (although I’m not totally sure because of the interaction terms).

Line edits

21 Like you do at 77, you should define “fall distance” here in the Abstract.

25 compound modifiers “long-distance migratory”, “short-distance migrants” need hyphens;

44 Extra period at end of sentence.

46 United States is abbreviated to US here but not in the Abstract at 21 and 22, where only the abbreviation is used without first defining it.

62 Suggest rewording “At the majority of US wind facilities that have been examined, bat mortality is estimated to be higher than bird mortality”.

95 Suggest replacing “unstandardized” with “non-standardized”.

115 Sentence is framed in past tense so it should be “allowed”

145 Insert “to” so it reads “we assigned records to the most…”

160 It’s not clear to me why there would be more than one global model? Or are the authors referring to the birds-only, bats-only, both global models?

199 “foliage-gleaning insectivores” missing hyphen

215 turkey vulture taxonomic name has not been provided.

225 hoary needs taxonomic name

247 Check the information for Reference 48. It is no longer referred to as The Birds of North America.

289 Insert hyphen “short-distance migrants”

290 Ditto “long-distance migrants”

325 Unclear what “Species were also comparable” means. Comparable to other studies?

327 “in the east” is much too casual. I assume you mean the eastern US?

328 Again, spell it out. “west coast of the US”.

333 Insert “of” so it reads “number of wind facilities”

334 Probably better to write “over the past 50 years”

337 Use past tense “carcasses were recovered”

342 Unclear what the 90% refers to? That these three species made up 90% of bat mortalities?

357 Or biased reporting? What the authors more generously refer to as “reporting culture” on L399.

369 Ambiguous. Differ from the present study or differ between large and small birds?

380 Seems odd to put the references in the middle of the sentence, given that the main message is that turbine height and blade length relate to fall distance. Suggest putting the reference at the end.

381-383 this seems redundant with 370-373.

391 I also think the authors should comment on how they assigned one value to all turbines at a facility. The fact that facility ID accounts for such high variation is important as it relates to variability in spatial arrangement and heights / designs of turbines at the facility. There could be particular areas of the facility that have higher mortality but that information is lost because the authors were not given turbine-specific mortality data. This is an important point and may show as much variation within a facility as there is among facilities. For example, topography within some facilities is likely to vary a lot whereas others may be more uniform in such factors.

413 It might be better to use “taxon-specific” rather than “taxa-specific”

452-462 This paragraph seems redundant with the previous paragraph. In particular, see 441-443.

467 What is a “curtailment regime”?

473-475 Not to mention weather patterns.

501 Suggest rewording as “assisted with data collection”

505 Reference seems incomplete. No place of publication, etc.

559 References in Annual Reviews do not typically indicate editors. They are more typically cited like journal articles.

561 Reference is incomplete. No publisher, place of publication.

567 Extra space on “wind-energy”

584 Journal name is not abbreviated or capitalized.

592 Is it correct that the paper is only one page?

593 Incomplete reference.

600 Check reference. Journal name not abbreviated, only one page listed.

618 Check style on title – every word is capitalized.

629 Reference is out of date. It is no longer referred to as Birds of North America.

646 Extra space on “information-theoretic”

661 Incomplete reference.

663 Incomplete reference.

667 Incomplete reference. No place of publication.

669 Ditto to above.

673 Ditto to above.

682 Italicize genera.

706 Is this a journal? One page? Please check.

734 Incomplete reference.

741 Incomplete reference.

744 Incomplete reference.

747 Incomplete reference.

Please check figure S2. When I downloaded it, it was blank.

The R code annotation is beautiful! Kudos to the second author.

7. PLOS authors have the option to publish the peer review history of their article (what does this mean?). If published, this will include your full peer review and any attached files.

Reviewer #1: No

Reviewer #2: No

Reviewer #3: No

---

## [Author Response · Author response to Decision Letter 1]

16 Jul 2020

Dear Dr. Magar,

My coauthors and I are grateful for the additional feedback provided by the reviewers, which has helped sharpen our manuscript. Enclosed is our second revision of our manuscript titled “An evaluation of bird and bat mortality at wind turbines in the Northeastern United States” (PONE-D-20-07129R1) that we wish to have considered for resubmission as an original research article in PLOS ONE. We believe that we have addressed the comments and concerns put forth by the reviewers, specifically those of Reviewer #3 related to our statistical analysis. For example, we have removed models with uninformative parameters. For some comments, we did not make changed in the manuscript but responded below, which we felt was appropriate. We have also added a few sentences to our discussion addressing points raised by Reviewer #3. Below, we respond to each reviewer’s comments individually. We have kept the original line numbers associated with each comment but in our response have included line numbers that reference the updated manuscript.

This resubmission represents original work that is not under consideration for publication elsewhere. Data in this manuscript are not included in previous publications nor will they be submitted for publication elsewhere. We are willing to cover page chargers and any other costs associated with the publication of the manuscript. This paper aims to make a contribution to the fields of conservation science, ornithology, mammalogy, and animal behavior, by reporting on patterns associated with bird and bat mortality at wind facilities, with an emphasis on the spatial arrangement of carcasses. To our knowledge, our dataset represents the largest ever compiled for bird and bat mortality at wind facilities in the Northeastern United States, an area where wind energy production is rapidly increasing. In addition, our empirical support for bird collisions with turbines poles in addition to blades is novel and germane for scientists, managers, and those involved in the wind power industry. 

Sincerely, 

Daniel Choi

Reviewer #1 General Comments

General Comment 1

The authors have done a thorough job of addressing my prior comments and I believe the manuscript is now basically ready for publication. I have only a few minor specific suggestions:

Reviewer #1 Specific Comments

Line 15: I suggest replacing “benefits to humans” with “environmental benefits” to help keep the manuscript focused and because I imagine there are some people who would disagree with this statement.

Response: We have made this change, and the sentence now reads “Wind energy offers substantial environmental benefits, but wind facilities can negatively impact wildlife, including birds and bats” (line 15). We had added “to humans” in response to a suggestion by Reviewer #3 during the first round of revisions. They suggested that we needed to specify benefits to humans, as opposed to wildlife. We hope that specifying “environmental benefits” finds the middle ground.

Line 59: I suggest replacing “Wind development” with something more clear and specific such as “Wind-energy development”.

Response: We have made this change and the sentence now reads “Wind energy development may have greater implications for the conservation of bats” (line 60). We did not use the hyphen, as others have not in this context. For example, see references 8, 9, 20, 26, 74, and 88.

Line 64: I suggest inserting “individuals” after “949,000”.

Response: Thank you for your suggestion. However, we have not made this change, because we specify earlier in the sentence that the numbers describe annual fatalities, and we think this is clear. The sentence reads “Estimate of annual bat fatalities in North America have ranged between 600,000 and 949,000, but similar to birds, the number of US turbines has increased substantially since these estimates were developed [14]” (lines 65-67).

Reviewer #2 General Comments

General Comment 1

I have now read the revised manuscript by Choi et al. and find it to be MUCH improved from the first iteration. I appreciate the author's very careful consideration of all reviewers. The introduction reads very clearly and I believe the new analysis approach clears up the results and they are easier for the reader to understand.

Response: Thank you for compliment and feedback.

Reviewer #2 Specific Comments

Line 170: Minor comment on Line 170: I think you mean < 6.

Response: Thank you for catching this error. We have changed “> 6” to “< 6”.

Reviewer #3 General Comments

General Comment 1

I thank the authors for submitting a much-improved manuscript. It is always encouraging when authors respond so diligently to reviewer suggestions / comments. In particular, I am glad to see that the stepwise analysis has been removed. That was obviously a deal breaker for me. I am satisfied with all of their responses to my edits / comments / queries. I do have some issues related to the revised manuscript, as follows:

Statistical Issues

1. Line 162: I believe Harrison’s method of looking at dispersion parameter > 1 only pertains to GLMs. The authors are using GLMMs so I would expect this to be the ratio of the sum of the squared Pearson residuals to the residual degrees of freedom, as outlined in Harrison (2014).

Response: Thank you identifying this issue. We have updated our calculation of the dispersion parameter using the correct ratio (lines 166-167).

2. Line 167: Many information theorists will cringe at the “all combinations” approach of constructing candidate models. Dochtermann and Jenkins provide some good commentary on why one should not dredge candidate models. Dochtermann, N. & Jenkins, S. (2011) Developing multiple hypotheses in behavioral ecology. Behavioral Ecology and Sociobiology, 65, 37-45.

Response: We appreciate you raising this concern, and we agree that an “all combinations” approach is generally best avoided. In constructing our global models, we made conscious effort to avoid including terms that had no basis in theory. For this reason, we omitted interactions such as mass:turbine height where we could not conceive of meaningful relationships between the variables. We think that all terms included in our global models, and subsequently entered into the MuMIn package’s dredge function, represent informed hypotheses about the potential relationships between turbine or species characteristic and fall distance. Our use of “all combinations” refers to all combinations of terms from the global models, which are limited to what we expected to be informative parameters, not “all combinations” of a large set of variables and their interactions, as cautioned against by Dotchermann and Jenkins (2011). Unfortunately, despite the connotations they may bring to some readers, we must use the phrase “all combinations” and reference the “dredge” function to represent our methodology accurately.

3. How confident are the authors in using package multimode? Has this package been vetted at all? The citation they give (reference 58) is to a preprint repository.

Response: The preprint status of this manuscript also gave us pause. However, the multimode package has recently been cited in articles published in International Statistical Review, Electronic Journal of Statistics, Animal Migration, Fungal Ecology, Journal of the Acoustical Society of America, and GigaScience. The package’s documentation is extremely thorough not only on methodology, but also on theory. Additionally, we found that the authors’ methodology builds upon established and well-cited statistical literature. Based on these facts, we are highly confident in the package.

4. Table 6 looks to me like it includes uninformative parameters. For example, the top model AM and the 4th model AHM have identical log likelihoods. In this case, H is likely an uninformative parameter. You can see this again when you compare models A vs AH (they differ by one parameter but the log likelihoods are identical). Again, H is an uninformative parameter. See also AD vs ADH. Please refer to Figure 1 in Leroux, S.J. (2019) On the prevalence of uninformative parameters in statistical models applying model selection in applied ecology. PLoS ONE, 14, e0206711. Please check all of your AIC tables for such uninformative parameters. I also see some in Table 7, and maybe in Table 8 (although I’m not totally sure because of the interaction terms).

Response Thank you for the suggestion. In following this guidance, we identified several uninformative parameters in our model sets. We have added screening for uninformative parameters to our methodology (lines 173-175) and updated our results (line 262, tables 6-9) accordingly. Furthermore, we have included a helper function in our supplementary R code that operates on MuMIn package model average objects to automate a significant portion of the screening process. We hope that this code may encourage future implementation of Leroux’s recommendations. We have kept models with uninformative variables in the tables following Arnold, T.W. (2010), Uninformative Parameters and Model Selection Using Akaike's Information Criterion. The Journal of Wildlife Management, 74: 1175-1178. doi:10.1111/j.1937-2817.2010.tb01236.x

Reviewer #3 Specific Comments

Line 21: Like you do at 77, you should define “fall distance” here in the Abstract.

Response: We have built this definition into the preceding sentence, which now reads “Understanding the horizontal fall distance between a carcass and the nearest turbine pole is important in designing effective search protocols and estimating total mortality” (lines 19-20).

Line 25: Compound modifiers “long-distance migratory”, “short-distance migrants” need hyphens;

Response: Thank you for pointing this out. We have added hyphens to this terms throughout.

Line 44: Extra period at end of sentence.

Response: We have removed the extra period.

Line 46: United States is abbreviated to US here but not in the Abstract at 21 and 22, where only the abbreviation is used without first defining it.

Response: We added “United States” to the sentence in the abstract, which now reads “We explored patterns in taxonomic composition and fall distance of bird and bat carcasses at wind facilities in the Northeastern United States using publicly available data and data submitted to the US Fish and Wildlife Service under scientific collecting and special purpose utility permits for collection and study of migratory birds” (lines 20-24). We additionally removed the use of the abbreviation “US” and “USFWS” in the abstract and instead define these upon first mention in the main body at lines 47 and 104. We did not change “US” in the title “US Fish and Wildlife Service,” as this is part of the official name of the agency.

Line 62: Suggest rewording “At the majority of US wind facilities that have been examined, bat mortality is estimated to be higher than bird mortality”.

Response: We made this change with small revisions: “At the majority of US wind facilities that have been examined, bat mortality has been estimated to be higher than bird mortality [2, 6]” (lines 63-65).

Line 95: Suggest replacing “unstandardized” with “non-standardized”.

Response: We have made this change.

Line 115: Sentence is framed in past tense so it should be “allowed”

Response: We have made this change.

Line 145: Insert “to” so it reads “we assigned records to the most…”

Response: We have inserted “to” before “records”, instead of after, as this would have changed the meaning. Added “each.” The sentence now reads “Where turbine dimensions varied within an individual facility, we assigned to each record the most common hub height and rotor diameter for that facility” (lines 145-147).

Line 160: It’s not clear to me why there would be more than one global model? Or are the authors referring to the birds-only, bats-only, both global models?

Response: We are referring to the global models and have inserted a parenthetical statement clarifying this: “Next, we measured the fit of each global model (birds-only, bats-only, and all-taxa global models) by using the MuMIn package [52] in R to calculate a marginal and conditional pseudo r-squared based on the Trigamma-estimate method” (lines 161-164).

Line 199: “foliage-gleaning insectivores” missing hyphen

Response: We have inserted a hyphen.

Line 215: turkey vulture taxonomic name has not been provided.

Response: We have added the taxonomic name for turkey vulture.

Line 225: hoary needs taxonomic name

Response: We have added the taxonomic name for hoary bat.

Line 247: Check the information for Reference 48. It is no longer referred to as The Birds of North America.

Response: Thank you for pointing this out. We have changed the reference here and in the references section.

Line 289: Insert hyphen “short-distance migrants”

Response: We have inserted a hyphen.

Line 290: Ditto “long-distance migrants”

Response: We have inserted a hyphen. 

Line 325: Unclear what “Species were also comparable” means. Comparable to other studies?

Response: We have added “to other studies.” The sentence now reads “Species were also comparable to other studies, predominantly red-eyed vireo, golden-crowned kinglet, and magnolia warbler” (lines 332-333).

Line 327: “in the east” is much too casual. I assume you mean the eastern US?

Response: We have added “ern US.” The sentence now reads “Raptors made up just 5% of fatalities, similar to other reports in the eastern US [34], but lower than raptor mortality along the west coast of the US [34, 66]” (lines 334-335).

Line 328: Again, spell it out. “west coast of the US”.

Response: We have specified the west coast, and the sentence now reads “Raptors made up just 5% of fatalities, similar to other reports in the eastern US [34], but lower than raptor mortality along the west coast of the US [34, 66]” (lines 334-335).

Line 333: Insert “of” so it reads “number of wind facilities”

Response: We have inserted “of”.

Line 334: Probably better to write “over the past 50 years”

Response: We have changed this to “over the past 50 years.”

Line 337: Use past tense “carcasses were recovered”

Response: We have changed to a past tense.

Line 342: Unclear what the 90% refers to? That these three species made up 90% of bat mortalities?

Response: We have added clarification in a parenthetical and the sentence now reads “Three migratory tree bats (hoary bat, silver-haired bat, and eastern red bat) were somewhat more common (90% of all bat fatalities in our data) than reported elsewhere [22, 68], but our data represent a slightly different geographic extent” (lines 349-351).

Line 357: Or biased reporting? What the authors more generously refer to as “reporting culture” on L399.

Response: Thank you for pointing this out. We have added “biased reporting” to this sentence, which now reads “This may be due to limited sample size, geographic differences, or biased reporting” (lines 365-366).

Line 369: Ambiguous. Differ from the present study or differ between large and small birds?

Response: We have clarified this sentence, which now reads: “Moreover, while AWWI’s [34] distance distributions did not appear to differ between large and small birds, this may have been the result of overly broad size bins” (lines 376-377).

Line 380: Seems odd to put the references in the middle of the sentence, given that the main message is that turbine height and blade length relate to fall distance. Suggest putting the reference at the end.

Response: Thank you for pointing this out. We have moved the reference to the end of the sentence.

Lines 381-383: this seems redundant with 370-373.

Response: Thank you for commenting on this. These sentences represent two different scenarios. With small turbines, larger birds are likely to fall farther from turbines. Similarly, but independently, larger turbines will have larger fall distances associated with them for all birds, regardless of size. Hence, we provide two different rationale for considering expansion of search areas/zones. As such, we feel the verbiage should remain as is. 

Line 391: I also think the authors should comment on how they assigned one value to all turbines at a facility. The fact that facility ID accounts for such high variation is important as it relates to variability in spatial arrangement and heights / designs of turbines at the facility. There could be particular areas of the facility that have higher mortality but that information is lost because the authors were not given turbine-specific mortality data. This is an important point and may show as much variation within a facility as there is among facilities. For example, topography within some facilities is likely to vary a lot whereas others may be more uniform in such factors.

Response: Thank you for suggesting this. We have added three additional sentences. First, we comment on the spatial arrangement of turbines within a facility: “For example, spatial patterns may be different at wind facilities with turbines arranged in a line (e.g., along a mountain ridge) than at those with turbines more evenly spaced in a group (e.g., in agricultural fields)” (lines 405-407). Second, we comment on how we used just one set of turbine dimensions per facility: “Variation between individual turbines within a wind facility may also affect spatial patterns of fall distance. For a few wind facilities which contained turbines of different sizes, we used only the most common turbine dimensions in our analysis, and this may have contributed to the variation accounted for by the random effect of facility ID” (lines 410-413).

Line 413: It might be better to use “taxon-specific” rather than “taxa-specific”

Response: We have made this change.

Lines 452-462: This paragraph seems redundant with the previous paragraph. In particular, see 441-443.

Response: Thank you for your comment. We think that these paragraphs are disparate enough to warrant inclusion of each. The first paragraph points to specific findings of our study and how the observed phenomenon could result in over or underestimation of mortality. We have modified the verbiage of the first paragraph in an effort to make this more clear (lines 455-459). The second paragraph is focused on the greater implications of collisions of birds with wind turbines poles.

Line 467: What is a “curtailment regime”?

Response: We have inserted a parenthetical definition, and the sentence now reads “Additionally, the distance distribution we present is based on various mortality survey procedures, surveyor expertise, curtailment regimes (purposeful reduction in turbine operation and electricity generation), and turbine models and sizes” (lines 480-483).

Lines 473-475: Not to mention weather patterns.

Response: Thank you for pointing this out. We have included mention of different weather patterns alongside our mention of geographic areas. The sentence now reads “For example, almost no turbines along the coast of the Great Lakes and Massachusetts are included in this dataset, and trends may be different for shorebirds or seabirds in certain geographic areas and areas with different weather patterns” (lines 488-490).

Line 501: Suggest rewording as “assisted with data collection”

Response: We have made this change.

Line 505: Reference seems incomplete. No place of publication, etc.

Response: Thank you for your attention to detail and for catching these many errors in our literature cited section. For this reference, no place of publication, other than the “United States,” is available, but we have added this. Further, no other information is available, but we have changed the standard link to a DOI link.

Line 559: References in Annual Reviews do not typically indicate editors. They are more typically cited like journal articles.

Response: We have removed the editors from the citation.

Line 561: Reference is incomplete. No publisher, place of publication.

Response: We have added this information.

Line 567: Extra space on “wind-energy”

Response: We have removed the extra space.

Line 584: Journal name is not abbreviated or capitalized.

Response: We have abbreviated and capitalized the journal name.

Line 592: Is it correct that the paper is only one page?

Response: We have filled in the correct page numbers.

Line 593: Incomplete reference.

Response: We have added the necessary information.

Line 600: Check reference. Journal name not abbreviated, only one page listed.

Response: We have added the correct page numbers. However, this is not a journal and does not have an accepted abbreviation.

Line 618: Check style on title – every word is capitalized.

Response: We have changed title to lowercase.

Line 629: Reference is out of date. It is no longer referred to as Birds of North America.

Response: We have updated the reference.

Line 646: Extra space on “information-theoretic”

Response: We have removed the extra space.

Line 661: Incomplete reference.

Response: We have updated this reference as per official PLOS ONE guidelines https://www.nlm.nih.gov/bsd/uniform_requirements.html

Line 663: Incomplete reference.

Response: We have updated this reference as per official PLOS ONE guidelines https://www.nlm.nih.gov/bsd/uniform_requirements.html

Line 667: Incomplete reference. No place of publication.

Response: We have added this information.

Line 669: Ditto to above.

Response: We have added this information.

Line 673: Ditto to above.

Response: We have added this information.

Line 682: Italicize genera.

Response: We have made this change.

Line 706: Is this a journal? One page? Please check.

Response: We have updated this with the correct reference from the Journal, Wind Energy Science.

Line 734: Incomplete reference.

Response: We have updated this reference.

Line 741: Incomplete reference.

Response: We have updated this reference.

Line 744: Incomplete reference.

Response: We have updated this reference.

Line 747: Incomplete reference.

Response: We have updated this reference.

Line 758: Please check figure S2. When I downloaded it, it was blank.

Response: We have re-uploaded this figure.

Line 833: The R code annotation is beautiful! Kudos to the second author.

Response: Thank you for your compliment.

---

## [Decision Letter · Decision Letter 2]

5 Aug 2020

PONE-D-20-07129R2

An evaluation of bird and bat mortality at wind turbines in the Northeastern United States

PLOS ONE

Dear Dr. Choi,

Thank you for submitting your manuscript to PLOS ONE. After careful consideration, we feel that the paper can be accepted after the authors address some few extra revisions as pointed out by the reviewer.

We look forward to receiving your revised manuscript.

Kind regards,

Vanesa Magar, Ph.D.

Academic Editor

PLOS ONE

Reviewers' comments:

Reviewer's Responses to Questions

**Comments to the Author**

1. If the authors have adequately addressed your comments raised in a previous round of review and you feel that this manuscript is now acceptable for publication, you may indicate that here to bypass the “Comments to the Author” section, enter your conflict of interest statement in the “Confidential to Editor” section, and submit your "Accept" recommendation.

Reviewer #3: (No Response)

2. Is the manuscript technically sound, and do the data support the conclusions?

Reviewer #3: Yes

3. Has the statistical analysis been performed appropriately and rigorously? 

Reviewer #3: Yes

4. Have the authors made all data underlying the findings in their manuscript fully available?

Reviewer #3: Yes

5. Is the manuscript presented in an intelligible fashion and written in standard English?

Reviewer #3: Yes

6. Review Comments to the Author

Reviewer #3: As before, the authors have done a great job at addressing my concerns and suggested edits. This manuscript will be an important contribution to the literature. It’s been a pleasure working with such attentive authors.

I have a few trivial line edits:

158 You are not really fitting models, you are ranking them.

174 Typo “be” should be “by”

196 I think you should use “0”, not “zero” in this context.

Table 3. I think it would be better to just report Total Population in millions. The extra zeros are clumsy.

411 Incorrect use of a restrictive clause. Change “which” to “that.

447 Extra period.

7. PLOS authors have the option to publish the peer review history of their article (what does this mean?). If published, this will include your full peer review and any attached files.

Reviewer #3: No

---

## [Author Response · Author response to Decision Letter 2]

5 Aug 2020

Dear Dr. Magar,

My coauthors and I are grateful for the additional feedback provided by the reviewers, which has helped sharpen our manuscript. Enclosed is our third revision of our manuscript titled “An evaluation of bird and bat mortality at wind turbines in the Northeastern United States” (PONE-D-20-07129R2) that we wish to have considered for resubmission as an original research article in PLOS ONE. We believe that we have addressed the brief comments put forth by Reviewer #3. We have also added two minor sentences to our discussion (lines 451-454) to mention a finding by the recent paper by Stokke et al. 2020, “Effect of tower base painting on willow ptarmigan collision rates with wind turbines,” which provides evidence for pole collisions by willow ptarmigan (Lagopus lagopus). Though this addition does not alter our discussion points in any way, we felt its inclusion was merited in our discussion of pole collisions and provided an important reference. Below, we respond to each reviewer’s comments individually. We have kept the original line numbers associated with each comment but in our response have included line numbers that reference the updated manuscript.

This resubmission represents original work that is not under consideration for publication elsewhere. Data in this manuscript are not included in previous publications nor will they be submitted for publication elsewhere. We are willing to cover page chargers and any other costs associated with the publication of the manuscript. This paper aims to make a contribution to the fields of conservation science, ornithology, mammalogy, and animal behavior, by reporting on patterns associated with bird and bat mortality at wind facilities, with an emphasis on the spatial arrangement of carcasses. To our knowledge, our dataset represents the largest ever compiled for bird and bat mortality at wind facilities in the Northeastern United States, an area where wind energy production is rapidly increasing. In addition, our empirical support for bird collisions with turbines poles in addition to blades is novel and germane for scientists, managers, and those involved in the wind power industry. 

Sincerely, 

Daniel Choi

Reviewer #3 Comments

General Comment 1

As before, the authors have done a great job at addressing my concerns and suggested edits. This manuscript will be an important contribution to the literature. It’s been a pleasure working with such attentive authors.

Response: We thank you for your continual investment in this paper and for reviewing each draft in such a thorough and timely manner.

I have a few trivial line edits:

Line 158: You are not really fitting models, you are ranking them.

Response: We have replaced “fitting” with “ranking.” The sentence now reads “Ranking models according to the Information-Theoretic approach [49], we first constructed global models containing all applicable predictor variables (Table 1)” (lines 158-159).

Line 174: Typo “be” should be “by”

Response: We have made this change.

Line 196: I think you should use “0”, not “zero” in this context.

Response: We have made this change.

Table 3: I think it would be better to just report Total Population in millions. The extra zeros are clumsy.

Response: Thank you for pointing this out. We have made this change. For example, instead of “130,000,000”, population numbers are listed as “130 million”. Instead of listing the column header as “Total Population (millions)”, we decided to include “million” after each value in the table, as we thought this created more ease for the reader.

Line 411: Incorrect use of a restrictive clause. Change “which” to “that.

Response: We have made this change.

Line 447: Extra period.

Response: We have removed the extra period

---

## [Editor Report · Decision Letter 3]

10 Aug 2020

An evaluation of bird and bat mortality at wind turbines in the Northeastern United States

PONE-D-20-07129R3

Dear Dr. Choi,

We’re pleased to inform you that your manuscript has been judged scientifically suitable for publication and will be formally accepted for publication once it meets all outstanding technical requirements.

Kind regards,

Vanesa Magar, Ph.D.

Academic Editor

PLOS ONE
---

## [Editor Report · Acceptance letter]

12 Aug 2020

PONE-D-20-07129R3 

An evaluation of bird and bat mortality at wind turbines in the Northeastern United States 

Dear Dr. Choi:

I'm pleased to inform you that your manuscript has been deemed suitable for publication in PLOS ONE. Congratulations! Your manuscript is now with our production department. 

Kind regards, 

on behalf of

Dr. Vanesa Magar 

Academic Editor

PLOS ONE